# FAM81A is a postsynaptic protein that regulates the condensation of postsynaptic proteins via liquid–liquid phase separation

Takeshi Kaizuka[1,2,3], Taisei Hirouchi[4], Takeo Saneyoshi[4], Toshihiko Shirafuji[2], Mark O. Collins[5,6], Seth G. N. Grant[3,7], Yasunori Hayashi[4], Toru Takumi[1,2,8]*

**1** RIKEN Brain Science Institute, Wako, Saitama, Japan, **2** Department of Physiology and Cell Biology, Kobe University School of Medicine, Kobe, Chuo, Kobe, Japan, **3** Centre for Clinical Brain Sciences, Chancellor's Building, Edinburgh BioQuarter, University of Edinburgh, Edinburgh, United Kingdom, **4** Department of Pharmacology, Kyoto University Graduate School of Medicine, Kyoto, Japan, **5** School of Biosciences, University of Sheffield, Sheffield, United Kingdom, **6** biOMICS Facility, Mass Spectrometry Centre, University of Sheffield, Sheffield, United Kingdom, **7** Simons Initiative for the Developing Brain (SIDB), Centre for Discovery Brain Sciences, University of Edinburgh, Edinburgh, United Kingdom, **8** RIKEN Center for Biosystems Dynamics Research, Chuo, Kobe, Japan

* takumit@med.kobe-u.ac.jp

**Data Availability Statement:** All relevant data are within the paper and its Supporting Information files.

## Abstract

Proteome analyses of the postsynaptic density (PSD), a proteinaceous specialization beneath the postsynaptic membrane of excitatory synapses, have identified several thousands of proteins. While proteins with predictable functions have been well studied, functionally uncharacterized proteins are mostly overlooked. In this study, we conducted a comprehensive meta-analysis of 35 PSD proteome datasets, encompassing a total of 5,869 proteins. Employing a ranking methodology, we identified 97 proteins that remain inadequately characterized. From this selection, we focused our detailed analysis on the highest-ranked protein, FAM81A. FAM81A interacts with PSD proteins, including PSD-95, SynGAP, and NMDA receptors, and promotes liquid–liquid phase separation of those proteins in cultured cells or in vitro. Down-regulation of FAM81A in cultured neurons causes a decrease in the size of PSD-95 puncta and the frequency of neuronal firing. Our findings suggest that FAM81A plays a crucial role in facilitating the interaction and assembly of proteins within the PSD, and its presence is important for maintaining normal synaptic function. Additionally, our methodology underscores the necessity for further characterization of numerous synaptic proteins that still lack comprehensive understanding.

## Introduction

Neurons communicate with each other through synapses to form a complex neuronal network. At the postsynaptic terminal of excitatory synapses, proteins involved in synaptic transmission and its regulation are highly enriched to form a postsynaptic density (PSD) [1–3]. Initially identified in electron microscopic analyses of synaptic structure, the relative resistance of PSD to detergent allowed biochemical isolation of the PSD fraction, which serves as a source

**Funding:** This work was supported by KAKENHI (JP16H06316, JP16H06463, JP21H04813, JP23H04233 to TT, JP18H05434, JP20K21462, JP22H04981 to YH, JP16J04376, JP18K14830 to TK) from the Japan Society for the Promotion of Science (JSPS) and the Ministry of Education, Culture, Sports, Science, and Technology (MEXT), Japan Agency for Medical Research and Development (AMED) under Grant Number JP21wm0425011 to TT, Japan Science and Technology Agency (JST) under Grant Number JPMJMS2299, JPMJMS229B to TT, Intramural Research Grant (30-9) for Neurological and Psychiatric Disorders of NCNP, the Takeda Science Foundation, Taiju Life Social Welfare Foundation to TT, and Human Frontier Science Program (RGP0020/2019) to YH. TK was supported by Grant-in-Aid for JSPS Fellows (JP16J04376). The funders had no role in study design, data collection and analysis, decision to publish, or preparation of the manuscript.

**Competing interests:** The authors have declared that no competing interests exist.

**Abbreviations:** BSA, bovine serum albumin; DIV, day in vitro; DMEM, Dulbecco's Modified Eagle Medium; FBS, fetal bovine serum; HBSS, Hanks' balanced salt solution; IDR, intrinsically disordered region; LFQ, label-free quantification; LLPS, liquid–liquid phase separation; MEA, multielectrode array; NGS, normal goat serum; NMDA, N-methyl-D-aspartate; PSD, postsynaptic density; SDS, sodium dodecyl sulfate; SDS-PAGE, sodium dodecyl sulfate-polyacrylamide gel electrophoresis; SMART, Simple Modular Architecture Research Tool.

for biochemical studies of the fraction [3–5]. Multiple proteomic analyses of the fraction have been conducted and uncovered a number of proteins involved in synaptic structure and function, including transmitter receptors, scaffolding proteins, cytoskeletal proteins, and signaling molecules [6–25]. Furthermore, yeast two-hybrid screening, immunoprecipitation, affinity purification, and proximity labeling using specific PSD components pulled out direct and indirect binding partners [26–34].

As the technologies of proteome analysis advance, the number of proteins detected in the PSD fraction increases and reaches the order of thousands [35]. Whereas many are bona fide PSD proteins from their known function and distribution, others are apparent contamination (such as proteins in glial cells or presynaptic compartments). In addition, multiple proteins without known functions or localization have been overlooked and left out of the in-depth analysis because it is difficult to determine whether they are contaminants or unreproducible.

Here, we performed meta-analyses of the multiple published datasets obtained under different experimental settings, assuming that those proteins identified in multiple datasets are authentic PSD proteins. We then identified and performed in-depth characterization of the top-ranked protein, FAM81A. Our data show that FAM81A is a functionally crucial postsynaptic molecule modulating the condensation of other PSD proteins. This approach of meta-analysis of multiple proteome datasets will help identify bona fide proteins in a sample that is otherwise overlooked from a single dataset.

## Results

### Meta-analysis of PSD proteome datasets

We carried out a meta-analysis of multiple sets of PSD proteome studies by quantitatively analyzing the combined results of the multiple investigations. We analyzed 35 datasets having at least 30 proteins (Table 1), of which 20 are from biochemically isolated PSD fractions followed by mass-spectrometric analysis (unbiased, Fig 1A) and 15 are by immunoprecipitation, pull-down, or proximal labeling of known PSD components (candidate-based, Fig 1B). A total of 5,869 proteins were detected at least in 1 dataset, where 5,800 proteins are in biochemical fractionation studies and 995 proteins in other studies (Fig 1C and S1 Table). Overall, the datasets from biochemical fractionation studies showed a high overlap of identified proteins, although they are derived from various samples that differed in species, brain region, and purification protocol. In contrast, immunoprecipitation, pull-down, or proximal labeling datasets showed relatively low overlap. The low overlap is observed even if the same starting protein (such as GluN2B or PSD-95) was used, suggesting the stochasticity of these approaches.

Among 5,869 proteins, about 4,000 proteins were detected in only 1–5 datasets (Fig 1D and S1 Data). These proteins may include contaminants from non-PSD proteins, as only 1.2% of them are known PSD proteins based on GO annotation (Fig 1E and S1 Data). In contrast, proteins detected reproducibly in multiple datasets contain a higher fraction of known PSD proteins. When we analyzed 123 proteins detected in more than 20 datasets, we found that about 40% of them are PSD proteins based on GO annotation, suggesting that PSD proteins are highly enriched in this group (Fig 1E and S1 Data). Analysis of the 123 proteins using SynGO showed enrichment of PSD proteins, as well as postsynaptic cytoskeleton and presynaptic proteins (Fig 1F). As for protein function, structural constituent of synapse and proteins related to trans-synaptic signaling are found to be enriched in them (Fig 1G). As expected, proteins detected in an even higher (>25) number of datasets and are likely more abundant proteins are composed of well-known core PSD proteins, including MAGUK family proteins and glutamate receptor subunits (S1 Fig and S1 Data). Our meta-analysis of PSD proteome datasets for

**Table 1. List of the 35 PSD proteome datasets referred to in this study.** No. 1–20 summarize the proteome data of biochemically purified PSD fraction (unbiased) and no. 21–35 are that of the PSD protein complex (candidate based). The original article, information on the sample and method, and the number of detected proteins of each dataset are described. Note that the number of proteins can be fewer than that shown in the original article because proteins that failed ID conversion were eliminated.

| No. | Publication | Type | Detail | Species | Note (sample) | Brain region | #Proteins | Converted |
|---|---|---|---|---|---|---|---|---|
| 1 | Jordan et al. Mol Cell Proteomics. 2004 3:857–71. | unbiased | PSD | Mouse, Rat | | Whole Brain | 306 | 286 |
| 2 | Collins et al. J Neurochem. 2006 97 Suppl 1:16–23. | unbiased | PSD | Mouse | | Forebrain | 620 | 620 |
| 3 | Trinidad et al. Mol Cell Proteomics. 2008 7:684–96. | unbiased | PSD | Mouse | | Cortex, Cerebellum, Midbrain, Hippocampus | 1090 | 741 |
| 4 | Nanavati et al. J Neurochem. 2011 119:617–29. | unbiased | PSD | Rat | | Hippocampus | 605 | 575 |
| 5 | Suzuki et al. J Neurochem. 2011 119:64–77. | unbiased | PSD | Rat | | Forebrain | 537 | 470 |
| 6 | Bayés et al. Nat Neurosci. 2011 14:19–21. | unbiased | PSD | Human | Biopsy | Neocortex | 1461 | 1453 |
| 7 | Bayés et al. PLoS One. 2012 7: e46683. | unbiased | PSD | Mouse | | Cortex | 1556 | 1554 |
| 8 | Distler et al. Proteomics. 2014 14:2607–13. | unbiased | PSD | Mouse | | Hippocampus | 2102 | 2102 |
| 9 | Han et al. Neuroscience. 2015 298:220–92. | unbiased | PSD | Rat | | Hippocampus | 1531 | 1496 |
| 10 | Föcking et al. Mol Psychiatry. 2015 20:424–32. | unbiased | PSD | Human | Postmortem | BA24; ACC | 727 | 705 |
| 11 | Bayés et al. Nat Commun. 2017 8:14613. | unbiased | PSD | Mouse | | Whole Brain | 2734 | 2731 |
| 12 | Bayés et al. Nat Commun. 2017 8:14613. | unbiased | PSD | Zebrafish | | Whole Brain | 2533 | 2023 |
| 13 | Föcking et al. Transl Psychiatry. 2016 6:e959. | unbiased | PSD | Human | Postmortem | BA24; ACC | 2033 | 2024 |
| 14 | Reim et al. Front Mol Neurosci. 2017 10:26. | unbiased | PSD | Mouse | | Hippocampus, Striatum | 2543 | 2543 |
| 15 | Roy et al. Nat Neurosci. 2018 21:130–138. | unbiased | PSD | Human | Postmortem | Neocortex (12 Brodmann Areas) | 1213 | 1210 |
| 16 | Roy et al. Proteomes. 2018 6. pii: E31. | unbiased | PSD | Mouse | | Cortex, Hippocampus, Hypothalamus, Striatum, Cerebellum | 1173 | 1173 |
| 17 | Dejanovic et al. Neuron. 2018 100:1322-1336.e7. | unbiased | PSD | Mouse | | Hippocampus | 1257 | 1257 |
| 18 | Wilson et al. Proteomes. 2019 7:12. | unbiased | PSD | Mouse | | Whole Brain | 2134 | 1896 |
| 19 | Kaizuka et al. bioRxiv 2022.05.05.490828v2 | unbiased | PSD | Mouse | | Whole Brain | 2186 | 2186 |
| 20 | Kaizuka et al. bioRxiv 2022.05.05.490828v2 | unbiased | PSD | Marmoset | | Neocortex, Hippocampus, Thalamus, Hypothalamus, Striatum, Cerebellum, Brainstem | 1916 | 1916 |
| 21 | Collins et al. J Neurochem. 2006 97 Suppl 1:16–23. | candidate based | GluN2B | Mouse | | Forebrain | 186 | 186 |
| 22 | Dosemeci et al. Mol Cell Proteomics. 2007 6:1749–60. | candidate based | PSD-95 | Rat | | Whole Brain | 288 | 283 |
| 23 | Fernández et al. Mol Syst Biol. 2009 5:269. | candidate based | PSD-95 | Mouse | | Forebrain | 119 | 119 |
| 24 | Schwenk et al. Neuron. 2012 74:621–33. | candidate based | GluA1-4 | Rat/ Mouse | | Whole Brain | 34 | 34 |
| 25 | Bayés et al. Mol Brain. 2014 7:88. | candidate based | GluN2B | Human | Biopsy | Frontal cortex | 239 | 238 |

(*Continued*)

**Table 1.** (Continued)

| No. | Publication | Type | Detail | Species | Note (sample) | Brain region | #Proteins | Converted |
|---|---|---|---|---|---|---|---|---|
| 26 | Bayés et al. Mol Brain. 2014 7:88. | candidate based | GluN2B | Human | Postmortem | Frontal cortex | 227 | 226 |
| 27 | Uezu et al. Science. 2016 353:1123–9. | candidate based | PSD-95 | Mouse | | Brain (Hippocampus/Cortex) | 121 | 121 |
| 28 | Loh et al. Cell. 2016 166:1295-1307.e21. | candidate based | LRRTM1/2 | Rat | | Cultured Neuron | 199 | 199 |
| 29 | Li et al. Nat Neurosci. 2017 20:1150–1161. | candidate based | PSD-95 | Mouse | Adult brain | Prefrontal Cortex | 133 | 133 |
| 30 | Li et al. Nat Neurosci. 2017 20:1150–1161. | candidate based | Dlgap1 | Mouse | Adult brain | Prefrontal Cortex | 139 | 139 |
| 31 | Li et al. Nat Neurosci. 2017 20:1150–1161. | candidate based | Shank3 | Mouse | Adult brain | Prefrontal Cortex | 107 | 107 |
| 32 | Li et al. Nat Neurosci. 2017 20:1150–1161. | candidate based | Cyfip1 | Mouse | | Prefrontal Cortex | 113 | 113 |
| 33 | Li et al. Nat Neurosci. 2017 20:1150–1161. | candidate based | Homer1 | Mouse | | Prefrontal Cortex | 35 | 35 |
| 34 | Li et al. Nat Neurosci. 2017 20:1150–1161. | candidate based | Syngap1 | Mouse | | Prefrontal Cortex | 77 | 77 |
| 35 | Li et al. Nat Neurosci. 2017 20:1150–1161. | candidate based | Tnik | Mouse | | Prefrontal Cortex | 81 | 81 |

the proteins recurrently detected in the multiple datasets successfully identifies known core PSD proteins.

## Identification of undercharacterized proteins from PSD proteome datasets

We asked whether this approach could detect proteins that have not been fully studied as PSD proteins. We first extracted protein names based on cloning ID or chromosome region (FamXX and XXXX...Rik) [36,37]. In addition, we also searched for proteins named after specific domains; Tmem for a transmembrane domain, Ccdc for a coiled-coil domain, Cctm for both of them, and Zfp for a zinc finger domain [38–42]. As a result, 177 proteins were identified from a total of 5,869 proteins (S2 Table). Among them, 97 proteins were detected in at least 2 datasets, and 21 proteins appeared in more than 8 datasets (Fig 2A and S1 Data). We focused on the top-ranked protein, FAM81A, which was detected in 21 datasets, including 15 PSD fractionations and 6 other studies (Fig 2A and 2B). Although a previous study confirmed the presence of FAM81A in the PSD using electron microscopy and immunoblotting [43], there are no studies of its structural or functional roles in the PSD.

## FAM81A is a higher vertebrate-specific PSD protein

Using protein BLAST, we found a single paralog of human FAM81A with 35% sequence identity, FAM81B. We searched for orthologs of FAM81A and FAM81B in vertebrate and invertebrate species using the protein BLAST and Gene2Function [44]. Both orthologs of FAM81A and FAM81B were found in mammals (mouse), birds (chicken), and reptiles (gecko) (Fig 2C), whereas only 1 ortholog (FAM81B) was found in amphibians (western clawed frog) and fish (zebrafish). As for invertebrates, we found only 1 FAM81 ortholog in tunicate (sea squirt), cephalochordate (lancelet), echinoderm (sea urchin), and mollusk (aplysia), whereas we could not find any FAM81 homologs in annelids, arthropod, and cnidaria (Fig 2C). This suggests that FAM81 duplicated during the evolution of higher vertebrates to yield 2 orthologs [45].

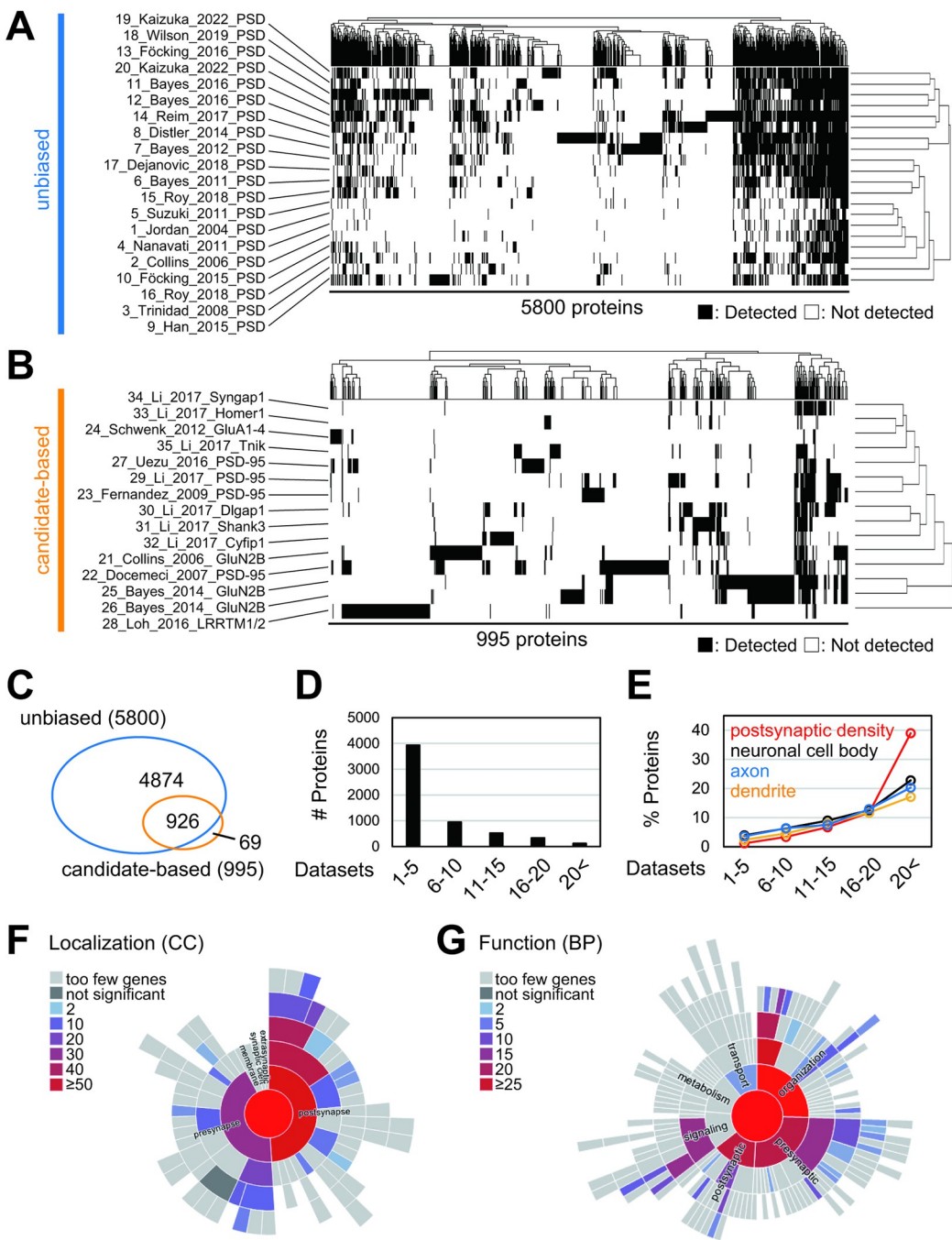

**Fig 1. Meta-analysis of the PSD proteome datasets.** (A and B) Overlap of the detected proteins in 35 PSD proteome datasets. "Protein complex" in the top panel indicates 995 proteins included in 15 datasets of the PSD protein complex. (C) Venn diagram describes the overlap of PSD fraction and PSD protein complex. (D) Histogram of the number of detected datasets. (E) The percentage of proteins that belong to indicated GO terms. (F and G) SynGO enrichment analysis of 123 proteins detected in more than 20 proteome datasets. The results of localization (Cellular Component; CC) (F) and function (Biological Process; BP) (G) are shown as -log10 Q-value. PSD, postsynaptic density.

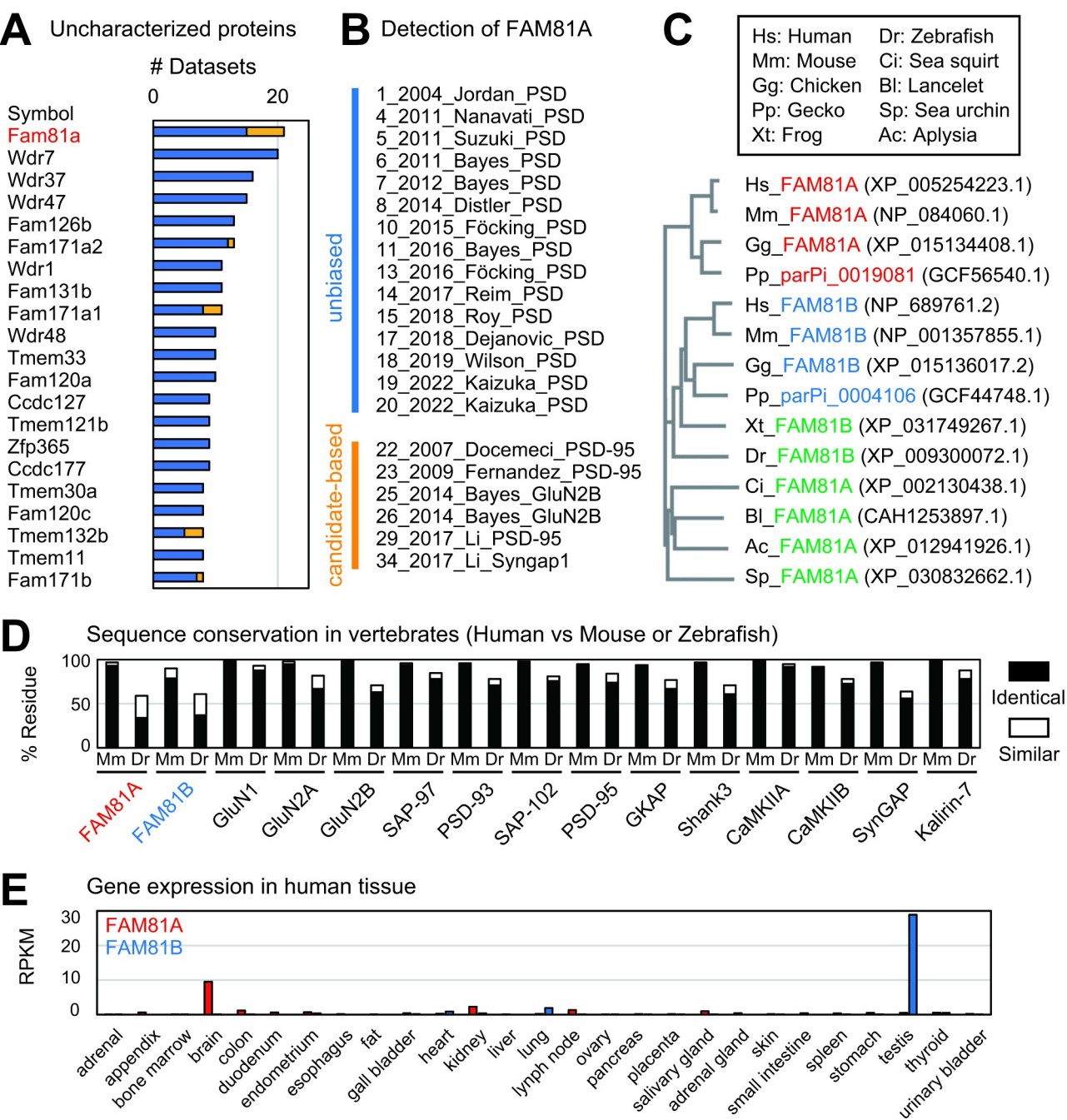

**Fig 2. FAM81A, a PSD protein expressed in the higher vertebrate brain.** (A) Hypothetical PSD proteins which are poorly characterized. Proteins identified in at least 8 datasets are listed. The blue and yellow bars indicate the dataset number of unbiased and candidate-based approaches. The name of proteins is described as the official symbol of the mouse gene. (B) Twenty-one PSD proteome datasets in which FAM81A (FAM81A) was identified. (C) Homologs of FAM81A. Red and blue items indicate orthologs of FAM81A and FAM81B, respectively. Green items indicate a common ortholog of FAM81A and 2 in amphibians, fish, and invertebrates. The phylogenetic tree was described with Kalign (https://www.ebi.ac.uk/Tools/msa/kalign/). Hs: *Homo sapiens*, Mm: *Mus musculus*, Gg: *Gallus gallus*, Pp: *Paroedura picta*, Xl: *Xenopus laevis*, Dr: *Danio rerio*. Ci: *Ciona intestinalis*, Bl: *Branchiostoma lanceolatum*, Sp: *Strongylocentrotus purpuratus*, Ac: *Aplysia californica*. (D) Expression pattern of FAM81A and FAM81 B in human tissues. The data was obtained from NCBI Gene. RPKM: Reads Per Kilobase of exon per Million mapped reads. (E) Sequence conservation of human FAM81A, FAM81B, and major PSD proteins in mouse (Mm) and zebrafish (Dr). The percentage of identical or similar residues was evaluated using Gene2Function. PSD, postsynaptic density.

Analysis of FAM81 protein domain architecture using the Simple Modular Architecture Research Tool (SMART) revealed that human FAM81A and FAM81B possess 2 and 4 coiled-coil domains, respectively (S2 Fig). In addition, we found that they contain regions predicted to be intrinsically disordered (S2 Fig).

Human FAM81A has more than 90% sequence identity with its mouse ortholog, which is comparable to other major PSD proteins (Fig 2D). By contrast, human FAM81A and FAM81B have only 34% and 37% sequence identity with zebrafish FAM81B, respectively, whereas other PSD proteins have 60% to 80% identity with their zebrafish orthologs (Fig 2D). These data suggest that FAM81A sequence evolved rapidly in the vertebrate lineage compared to other PSD proteins.

Analysis of mRNA expression shows FAM81A and FAM81B are expressed exclusively in the brain and testis, respectively, in humans and mice (Figs 2E and S3A, and S1 Data). In contrast, in amphibians FAM81B is expressed in a broad range of tissue, suggesting that the regulation of FAM81A and FAM81B expression has differentiated during vertebrate evolution (S3B Fig and S1 Data).

## FAM81A is distributed on PSD and partially colocalized with NMDA receptor

To confirm the previous studies showing FAM81A localizes at the PSD, we analyzed protein expression in synaptic extracts. We found that FAM81A was enriched in the detergent-resistant PSD fraction, similarly to the bona fide PSD protein, PSD-95, whereas a presynaptic protein, synaptophysin, was depleted from the fraction, consistent with the previous report (Fig 3A) [43]. Immunoprecipitation of PSD-95 using crude synaptosome fraction showed interaction of FAM81A with PSD-95, confirming the proteome results (Fig 3B). We then examined the localization pattern of FAM81A protein in mouse brain using immunohistochemistry. The signal of FAM81A was observed across most regions of the brain (Fig 3C). High magnification imaging of FAM81A immunostained brain sections showed it was found to form punctate structures in the adult brain, similar to other synaptic proteins (Fig 3D). The punctate structure of FAM81A was also observed at postnatal day 7 when PSD-95 is expressed at very low levels (S4A and S4B Fig) [34,46]. This suggests that the PSD localization of FAM81A is independent of PSD-95. As FAM81A has been detected in protein complexes of N-methyl-D-aspartate (NMDA) receptor [47], we wondered whether FAM81A colocalizes with the NMDA receptor. Co-staining of FAM81A and GluN1 subunit of NMDA receptor showed partial overlap between those puncta (Fig 3E), indicating that some synapses express either or both of these proteins.

To evaluate the relative abundance of FAM81A protein in synapses, we reanalyzed MS data from our previous proteomic study of mouse synaptosome and PSD fractions [19]. We estimated the relative abundance of proteins in this dataset using iBAQ intensities and found that although FAM81A is not very abundant in mouse synaptosomes (abundance rank 2,674 out of 3,818 proteins), the abundance of FAM81A is higher in PSDs (abundance rank 304 out of 3,314 proteins) similarly to the enrichment of other bona fide PSD proteins in PSD preparations compared to synaptosomes (Fig 3F and 3G and S1 Data). Taken together, these data indicate that FAM81A is a major postsynaptic protein that partially colocalizes with NMDA receptors.

## FAM81A forms condensate in neurons

To analyze intracellular localization and dynamics of FAM81A, we next observed FAM81A in cultured neurons. Exogenously expressed FAM81A-GFP and PSD-95-mCherry were colocalized at the tip of dendritic spines, suggesting the PSD localization of FAM81A in primary

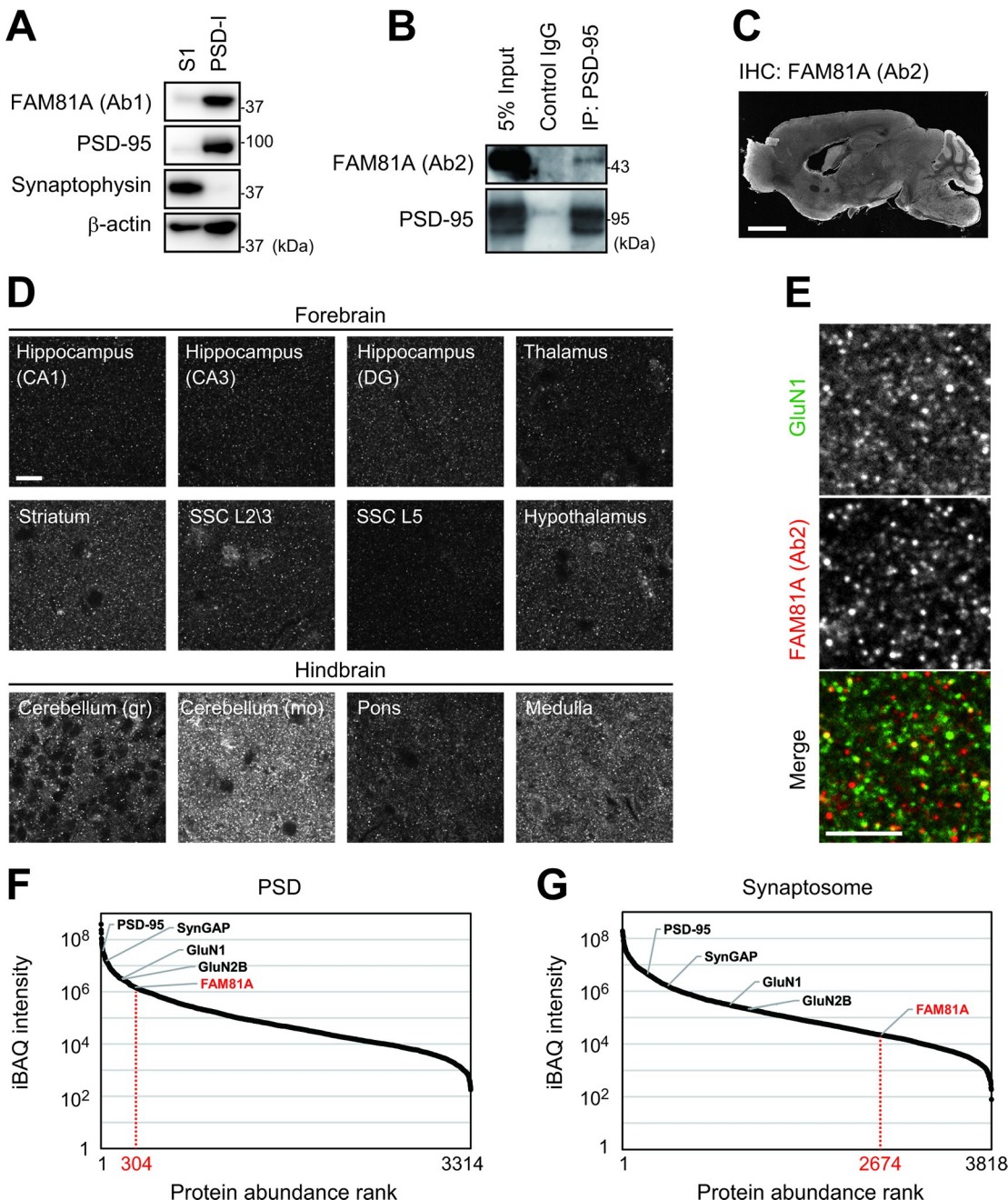

**Fig 3. FAM81A is a PSD protein that partially colocalizes with NMDA receptor.** (A) Enrichment of FAM81A in PSD. S1 fraction and PSD-I fraction were prepared from brain homogenate of adult mice. (B) Interaction of FAM81A with PSD-95 in synapse. Crude synaptosome fraction of adult mouse forebrain was subjected to immunoprecipitation using anti-PSD-95 antibody. (C and D) Distribution of FAM81A in the brain. Immunohistochemistry of FAM81A was performed on the sagittal section of the adult mouse brain. (E) Colocalization of FAM81A and NMDA receptor. Immunohistochemistry of FAM81A and GluN1 was performed on the section of the adult mouse brain. The bottom part of the dentate gyrus is described with high magnification. (F and G) Relative abundance of individual proteins in mouse PSD (F) and synaptosome fractions (G) was estimated using iBAQ intensities calculated from our published proteomic data [19]. Scale bar: 2 mm (C), 10 μm (D), or 5 μm (E). NMDA, N-methyl-D-aspartate; PSD, postsynaptic density.

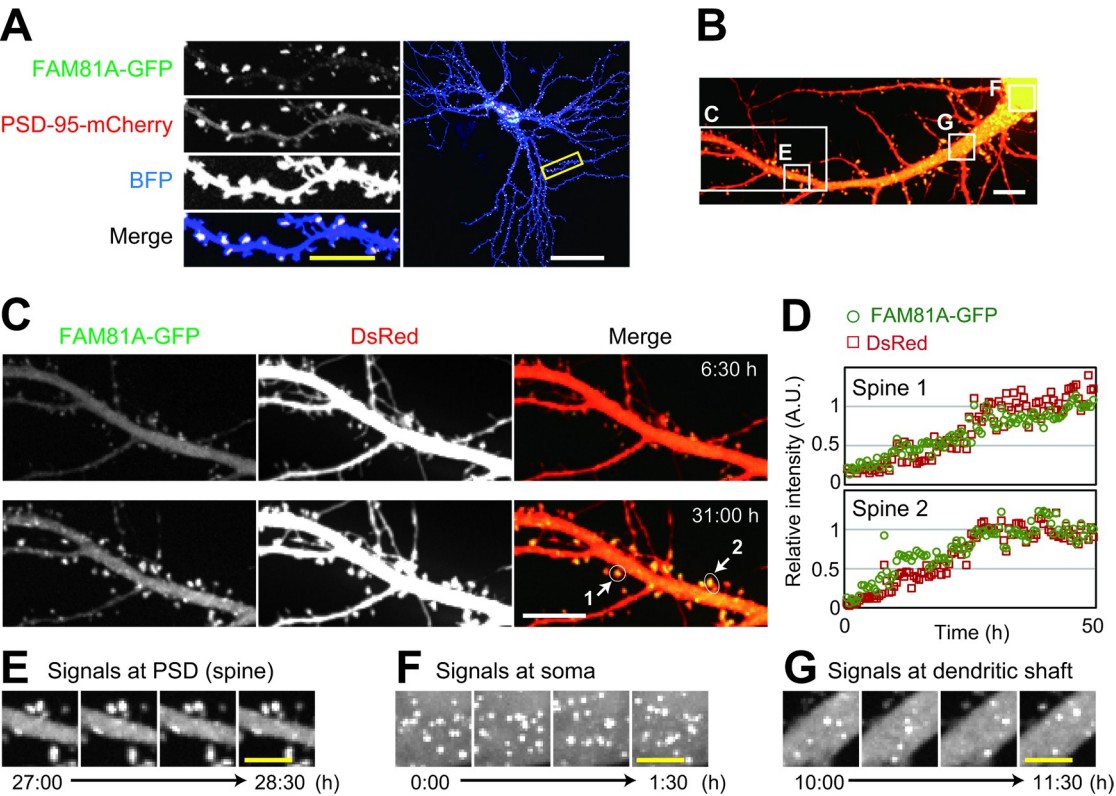

**Fig 4. Formation of FAM81A condensates at PSD and cytoplasm in neuron.** (A) Localization of FAM81A at PSD in the cultured neuron. Primary mouse hippocampal neurons were transfected with FAM81A-GFP, PSD-95-mCherry, and BFP at DIV19. Two days later, the cells were fixed and observed with a confocal microscope. (B–G) Live-imaging of primary cultured mouse hippocampal neurons (DIV16-18) transfected with FAM81A-GFP and DsRed (30 min/frame). (B) Overview of the observed neuron. Regions magnified in panels C, E, F, and G are described. See also S1 Movie. (C and D) Accumulation of FAM81A on the PSD upon spine maturation. The signal intensity of 2 spines shown in panel C is quantified and plotted in panel D. (E) Puncta of FAM81A-GFP on the PSD. (F) Puncta of FAM81A-GFP at soma. (G) Puncta of FAM81A-GFP at dendritic shaft. Scale bars: (A) 50 μm (white) or 10 μm (yellow). (B–G) 10 μm (white) or 5 μm (yellow). DIV, day in vitro; PSD, postsynaptic density.

hippocampal neurons (Fig 4A). We then performed time-lapse imaging of FAM81A-GFP expressed in hippocampal neurons from DIV (day in vitro) 16 to 18 (Fig 4B and S1 Movie). As the neuron matures and gains mushroom-shaped dendritic spines, FAM81A was condensed in the structure (Fig 4C and 4D and S1 Data). In addition to accumulation at dendritic spines, we found the condensates of FAM81A in the soma and dendritic shafts (Fig 4E–4G). These cytoplasmic condensations do not remain in the same position for 30 min, in contrast to stable localization of the puncta on PSD (Fig 4E–4G). These results suggest that FAM81A not only accumulates on PSD but also forms dynamic condensates in the cytoplasm.

## Domain structure of FAM81A required for condensation

We attempted to reproduce the condensate in a heterologous system and found that it can also be reproduced in HEK293T heterologous expression systems, where synaptic proteins are hardly expressed. This suggests that FAM81A has a propensity to condense without other synaptic molecules (Fig 5A and S2 Movie). To examine the molecular architecture of FAM81A important for the condensation, we asked for the sequence motif required for condensation by generating a series of deletion mutants and expressing them in HEK293T cells (Figs 5B and S5). Expression of the GFP tagged mutants was confirmed with immunoblotting (S6A Fig).

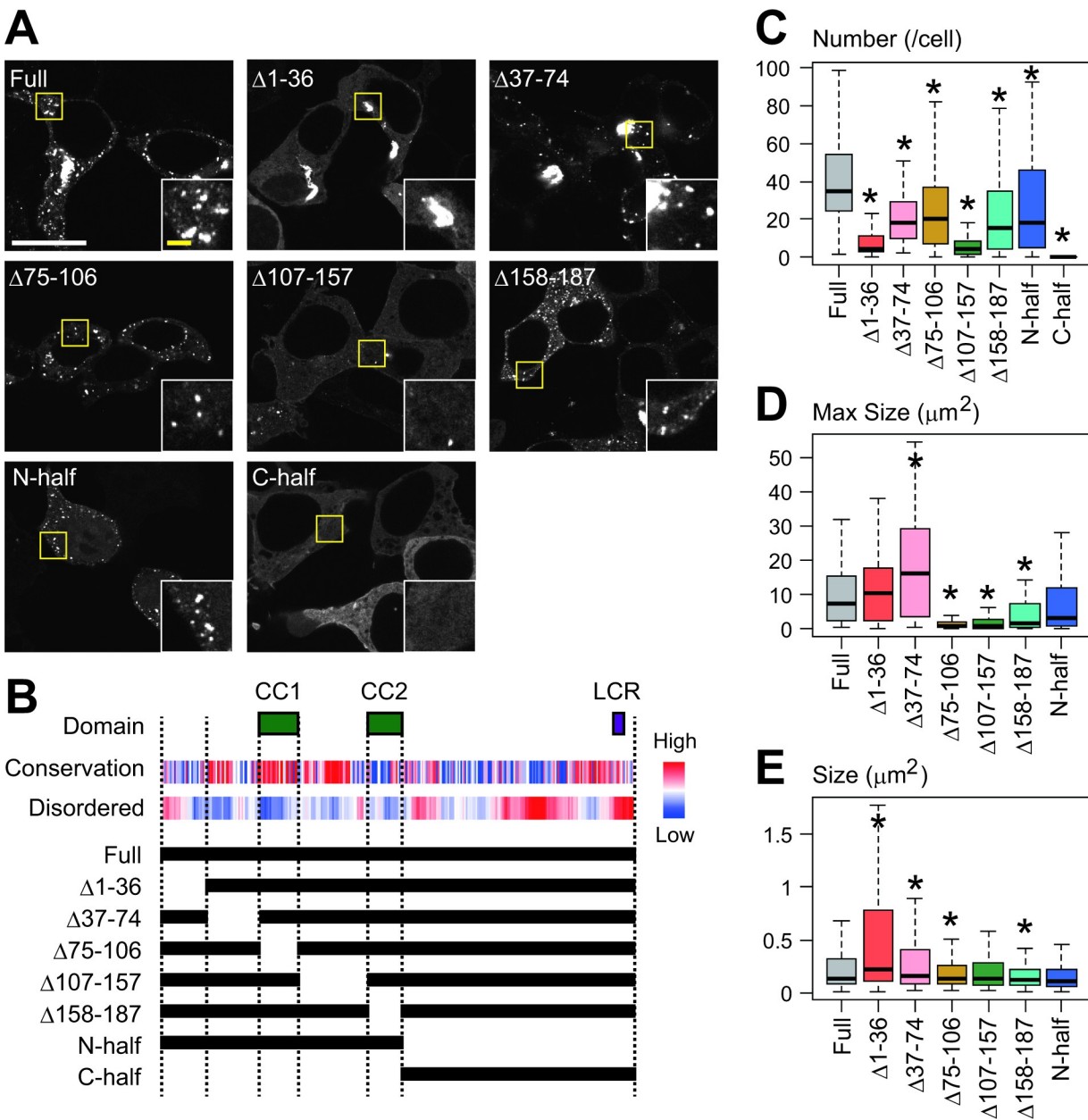

**Fig 5. Domain structure of FAM81A required for condensation.** (A) Condensate formation of FAM81A mutants. HEK293T cells were transfected with indicated FAM81A mutants, and 24 h later, the cells were fixed and observed with a confocal microscope. (B) Primary structure of mouse FAM81A. The coiled-coil domains (CC1 and CC2) and LCR were identified using SMART. Evolutional conservation was analyzed using ConSurf. The probability of disorder is assessed using IUPred2A. Truncate mutants described at the bottom were used in (A). (C–E) Quantitative analysis of FAM81A droplets described in (A). Number of FAM81A condensates per cell (C) and size of FAM81A condensates. (D) Maximum size of FAM81A condensates in individual cells (E) are described. The numbers of analyzed cells expressing full length, Δ1–36, Δ37–74, Δ75–106, Δ107–157, Δ158–187, N-half, or C-half FAM81A are 98, 85, 87, 109, 95, 101, 93, or 83, respectively. Scale bars: 50 μm (white) or 10 μm (yellow). *$P < 0.05$. LCR, low complexity region; SMART, Simple Modular Architecture Research Tool.

Although the expression level of most mutants was comparable to that of full-length FAM81A, C-terminal half (C-half; 188–364) showed a significantly high expression level (S6A and S6B Fig and S1 Data). Thus, we analyzed cells expressing these mutants at similar expression levels (see Methods). We found that the C-half completely abolished the puncta formation, whereas

the N-terminal half (N-half; 1–187) still formed the puncta, indicating that the N-terminal half is essential for condensation (Fig 5A and 5C and S1 Data). Analysis of smaller deletion mutants of the N-terminal half showed that all of these mutants resulted in decreased puncta formation (Fig 5A and 5C and S1 Data). In particular, Δ1–36 and Δ107–157 showed an approximately 90% decrease in the number of condensates. The size and/or maximum size of puncta of the Δ1–36 and Δ37–74 mutants were larger and more amorphous than that of full-length FAM81A, suggesting that these mutants form aggregates (Fig 5A, 5D and 5E and S1 Data). Δ75–106, Δ107–157, and Δ158–187 showed a defect in the formation of enlarged structures and the size of Δ75-106- and Δ158-187-positive structures were smaller than that of full-length FAM81A, suggesting that these regions contribute to the formation of FAM81A droplets. Taken together, these results indicate that the N-terminus (1–36) and the sequence between 2 coiled-coil domains (107–157) of FAM81A are important for its condensation, and the N-terminal region (1–74) is important to avoid aggregate formation.

## Condensation-mediated localization of FAM81A enlarges dendritic spines

To test if condensation is required for synaptic accumulation, we observed the localization of condensation-deficient FAM81A mutant Δ107–157 expressed in the hippocampal neuron. It showed diffused cytosolic localization and accumulation at PSD was significantly impaired compared with the full-length FAM81A, suggesting that condensation of FAM81A is essential for its PSD localization (Fig 6A). We wondered whether PSD localization of FAM81A affects the size of the dendritic spine. As a result, the size of dendritic spines on neurons expressing full-length FAM81A was larger than that of neurons expressing Δ107–157 mutant (Fig 6B and S1 Data). This suggests that accumulation of FAM81A on PSD enlarges spine size. To test whether condensate formation and PSD localization of FAM81A is mediated by interaction with PSD-95, we tested whether FAM81A mutants defective in condensate formation are also defective in PSD-95 binding. We found that PSD-95-GFP can be co-precipitated with FAM81A-FLAG when overexpressed in HEK293T cells (Fig 6C). Unexpectedly, we found that all deletion mutants tested here, including Δ107–157 co-precipitate PSD-95-GFP in similar extent, suggesting that FAM81A has multiple binding sites for PSD-95 (Fig 6C). The interaction of Δ107–157 mutant with PSD-95 indicates that binding with PSD-95 is not sufficient for its condensation and PSD localization. Together, these results indicate that localization of FAM81A to PSD needs its condensation, and accumulation of FAM81A on PSD enlarges dendritic spines.

## FAM81A forms condensate by liquid–liquid phase separation

To uncover the mechanism of FAM81A condensate formation, we performed a more detailed observation of the condensates using time-lapse imaging. We found that FAM81A puncta showed flexible shapes and underwent fusion and fission (Fig 7A–7C), indicating that these punctate FAM81A structures have liquid-like properties rather than solid aggregates. We also observed larger structures, often with complicated shapes (Fig 7D). The shape of such structures was stable over time, suggesting that they are rigid protein aggregates (Fig 7D and S3 Movie). These observations suggest that FAM81A forms these condensates through the mechanism of liquid–liquid phase separation (LLPS). To test this, we examined the effect of 1,6-hexanediol that disrupts LLPS by interfering with hydrophobic interactions [48] on FAM81A droplets in HEK293T cells. Although large aggregate-like structures remained, most of the puncta disappeared after 10 min incubation, consistent with the hypothesis that FAM81A undergoes LLPS (Fig 7E). We then tested whether the multimerization of FAM81A is involved in LLPS. We found that GFP-FAM81A was co-precipitated by FAM81A-FLAG expressed in

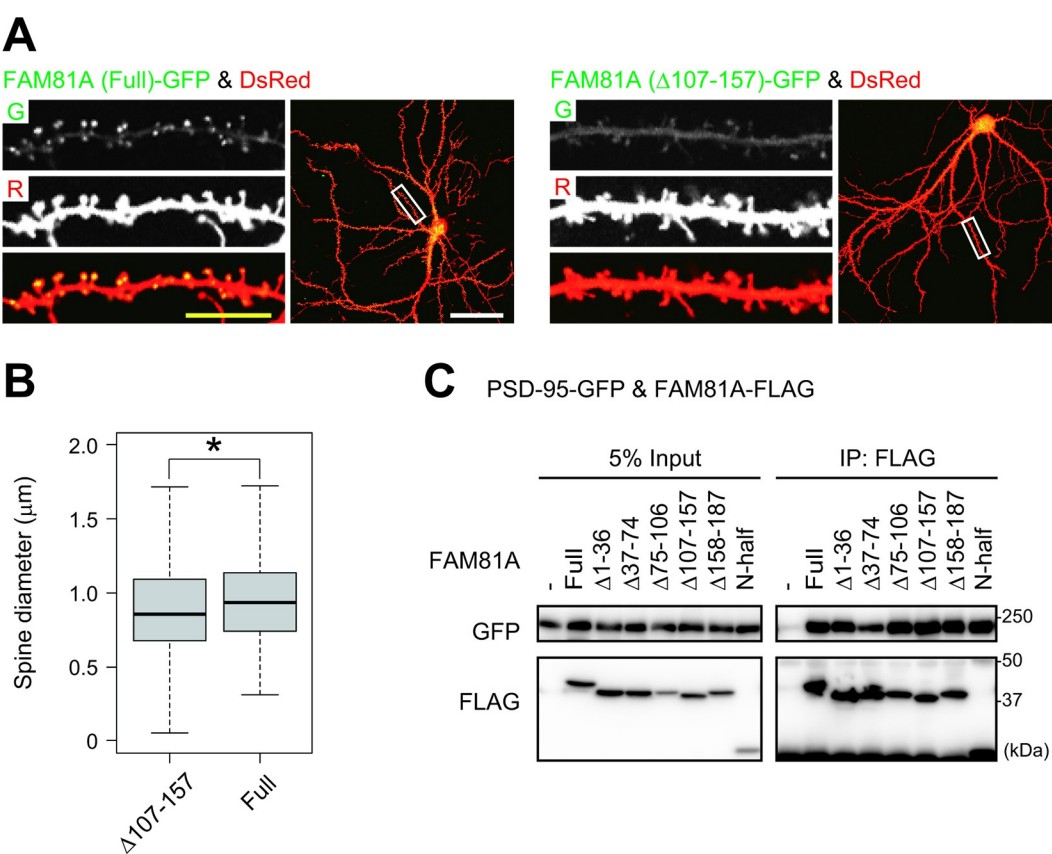

**Fig 6. Condensation-mediated localization of FAM81A on PSD enlarges dendritic spines.** (A) Impaired PSD localization of the condensation-deficient FAM81A mutant. Primary cultured mouse hippocampal neurons were transfected with full-length or Δ107–157 FAM81A mutants together with DsRed at DIV19. Two days later, the cells were fixed and observed with a confocal microscope. (B) Enlargement of dendritic spines by accumulation of FAM81A on PSD. The diameter of 1,337 dendritic spines on neurons expressing Δ75–106 FAM81A and 560 spines on neurons expressing full-length FAM81A were quantified. (C) Interaction of FAM81A mutants with PSD-95. HEK293T cells were transfected with PSD-95-GFP and indicated FAM81A mutants tagged with FLAG; 24 h later, cells were lysed, and immunoprecipitation was performed with an anti-FLAG antibody. Then, immunoblotting was performed with anti-GFP or anti-FLAG antibodies. Scale bars: 50 μm (white) or 10 μm (yellow). *$P < 0.05$. DIV, day in vitro; PSD, postsynaptic density.

HEK293T cells, suggesting that FAM81A multimerizes with each other, to drive its LLPS (Fig 7F).

## FAM81A interacts and forms condensate with core synaptic proteins

We then tested if FAM81A can interact with other major PSD proteins. We found that GFP-SynGAP and GFP-GluN2B expressed in HEK293T cells can be co-immunoprecipitated with FAM81A-FLAG, as well as PSD-95-GFP (Fig 7G). The interaction of FAM81A with these proteins in non-neuronal cells, which hardly expresses synaptic molecules, suggests that FAM81A directly binds to PSD-95, SynGAP, or GluN2B.

It has been shown that PSD-95 and SynGAP undergo LLPS when they are combined [49,50]. Consistently, we reproduced the condensation of GFP-SynGAP and PSD-95-mCherry in HEK293T cells upon co-expression (S7 Fig). Given the interaction of FAM81A with both PSD-95 and SynGAP, we tested whether FAM81A can undergo LLPS with SynGAP or PSD-95 in HEK293T cells. As a result, GFP-SynGAP and FAM81A-FLAG condensed together (Fig 7H). On the other hand, PSD-95-mCherry co-expressed with FAM81A-GFP showed diffuse

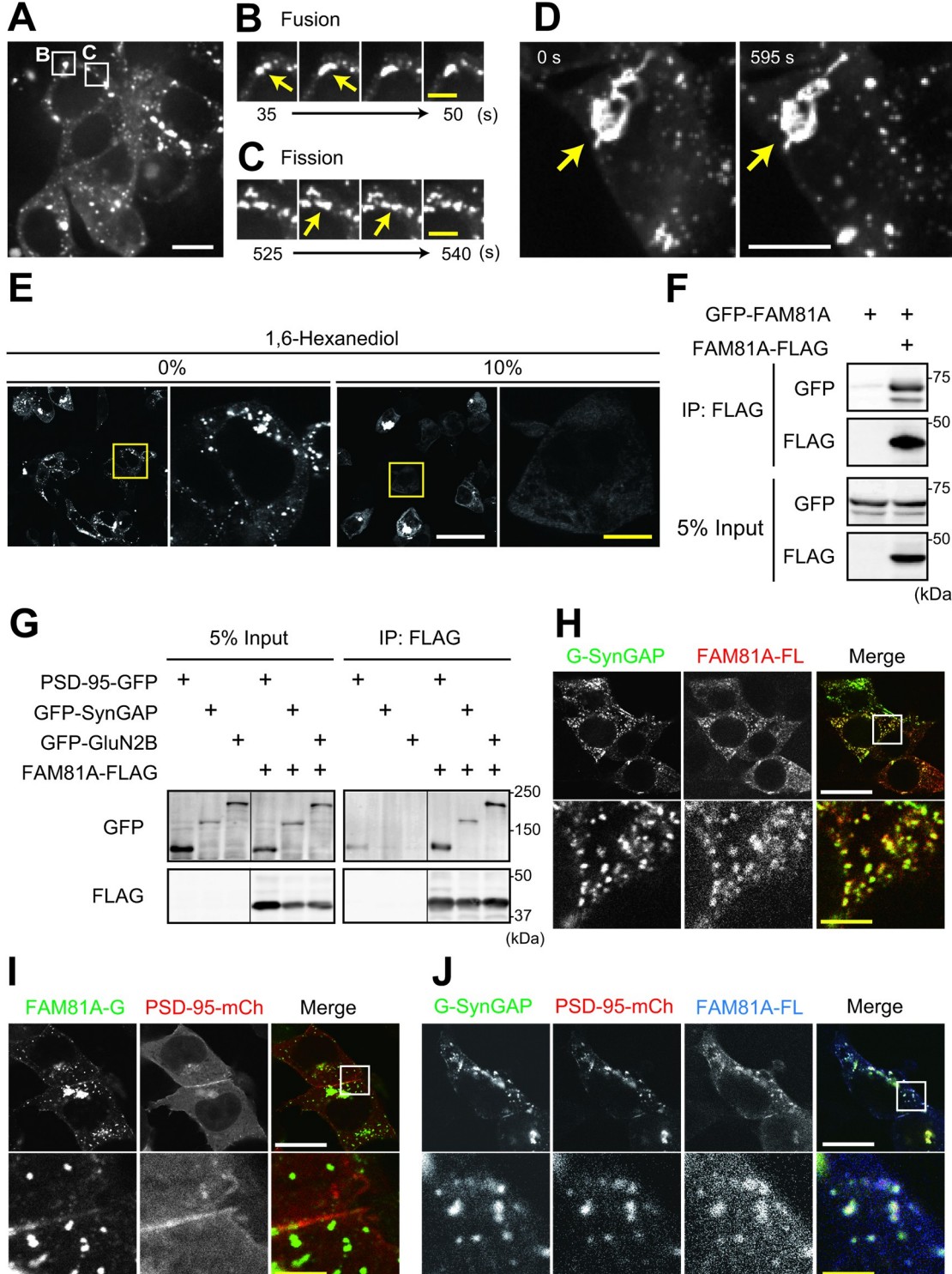

**Fig 7. Interaction and co-localization of FAM81A with core synaptic molecules.** (A–D) Live-imaging of HEK293T cells transfected with FAM81A-GFP. (A) Overview of the observed cells. Regions magnified in panels B and C are described. See also S2 Movie. (B and C) Representative movie images of punctate structures undergoing fusion (B) and fission (C). (D) Enlarged rigid structure with a stable shape. See also S3 Movie. (E) The effect of 1,6-hexanediol on FAM81A positive structures. HEK293T cells were transfected with FAM81A-GFP, and 24 h later, the medium was changed to fresh medium, including the indicated concentration of 1,6-hexanediol. Ten min later, the cells were fixed. (F and G) Interaction between FAM81A or interaction of FAM81A and PSD-95, SynGAP, or GluN2B. HEK293T cells were transfected with indicated plasmids, and 24 h later, cells were

lysed, and immunoprecipitation was performed with an anti-FLAG antibody. Then, immunoblotting was performed with anti-GFP or anti-FLAG antibodies. (H–J) Localization of FAM81A on SynGAP-positive droplets. HEK293T cells were transfected with indicated plasmids, and 24 h later, cells were fixed and observed with confocal microscopy. For panels H and J, cells were subjected to immunocytochemistry using an anti-FLAG antibody before observation. Scale bars: 10 μm (white) or 5 μm (yellow) in (A–D), 50 μm (white) or 10 μm (yellow) in (E), and 20 μm (white) or 4 μm (yellow) in (H–J), respectively. PSD, postsynaptic density.

distribution, hardly localized with the punctate distribution of FAM81A-GFP (Fig 7I). However, when all 3 were co-expressed, they formed condensate together, suggesting that PSD-95 needs SynGAP for condensation with FAM81A (Fig 7J). These results indicate that FAM81A interacts with core synaptic proteins and co-localizes with SynGAP-positive droplets.

To test if FAM81A can undergo LLPS along with PSD-95, GluN2B, and SynGAP in vitro, we bacterially expressed and purified these proteins, combined, and observed them under a microscope. As a result, FAM81A formed condensate in combination with SynGAP, GluN2B, and PSD-95 (3 μM each) (Fig 8A). To examine the role of FAM81A in forming condensate, we next decreased the concentration of FAM81A to 1 or 0 μM while maintaining the concentration of other proteins. Upon reducing the concentration of FAM81A, we found a decrease in the condensate size, as visualized in the PSD-95 channel (Fig 8A and 8B and S1 Data). This indicates that FAM81A can facilitate the condensate formation of PSD proteins through the assembling and stabilizing the component proteins.

## FAM81A affects PSD size and neuronal activity

To examine the role of FAM81A on the formation of PSD in neurons, we performed a knockdown experiment of FAM81A in the cultured hippocampal neuron. We expressed 2 different shRNAs against FAM81A (shFAM81A #1 and #2) by using a lentivirus vector, both of which down-regulated the mRNA level of FAM81A to <10% (S8A Fig and S1 Data). We then analyzed PSD-95 puncta on neurons, using GFP to visualize the dendrites of the infected neurons (S8B Fig). The size of PSD-95 puncta is decreased in neurons expressing shRNA for FAM81A (Fig 9A and 9B and S1 Data), suggesting that FAM81A stabilizes PSD-95 at the synapse. To test the physiological role of FAM81A, we next examined whether the neuronal activity is affected by FAM81A down-regulation using a multielectrode array (MEA). We found a significant decrease in the frequency of neuronal firing in FAM81A down-regulated neurons, indicative of reduced excitatory synaptic transmission (Fig 9C and 9D and S1 Data). These results suggest the structural and functional importance of FAM81A at the excitatory synapse.

## Discussion

The proteome of the PSD has a high complexity comprising thousands of proteins and characterizing their structural and functional interactions is essential for understanding how the complexity of the PSD ultimately controls neuronal networks. Toward understanding those proteins in the PSD that remain poorly characterized, we developed a meta-analysis approach that identified a set of 97 proteins of which FAM81A was a prominent member. In-depth analysis of FAM81A expression, protein interactions and condensation properties, and electrophysiological role indicate that FAM81A is important for synaptic function and maintenance of neuronal networks.

The FAM81 gene family in mammals comprises 2 paralogs, FAM81A and FAM81B, whereas fish, amphibians, and some invertebrate genomes contain a single FAM81 gene. The most parsimonious explanation is that FAM81 underwent a gene duplication in the branch of the vertebrate lineage leading to mammals approximately 300 to 100 million years ago. Following the duplication event, FAM81A and FAM81B paralogs evolved distinct tissue expression

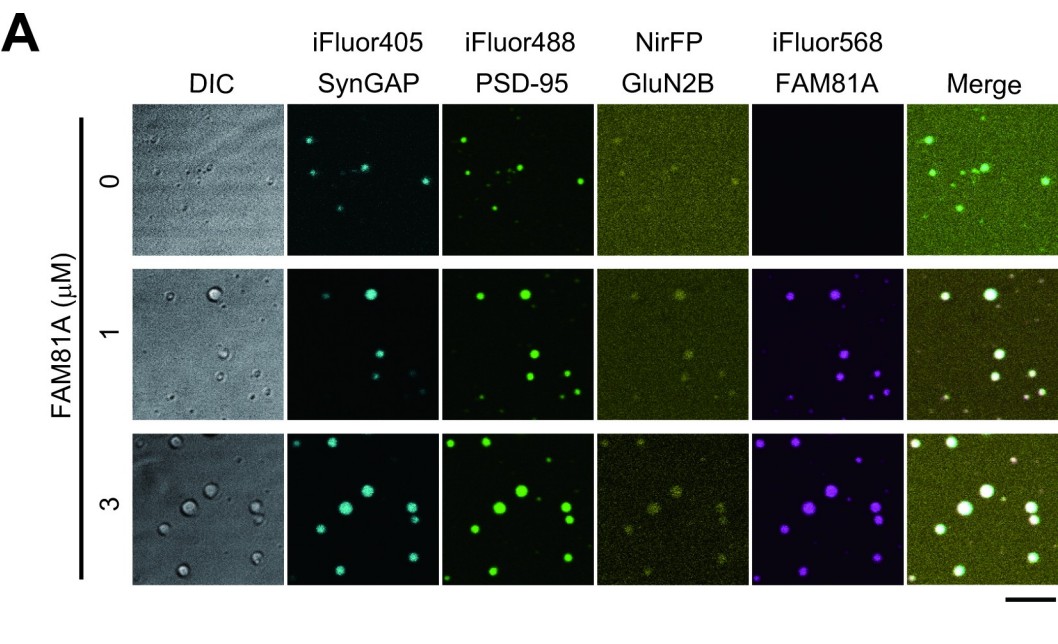

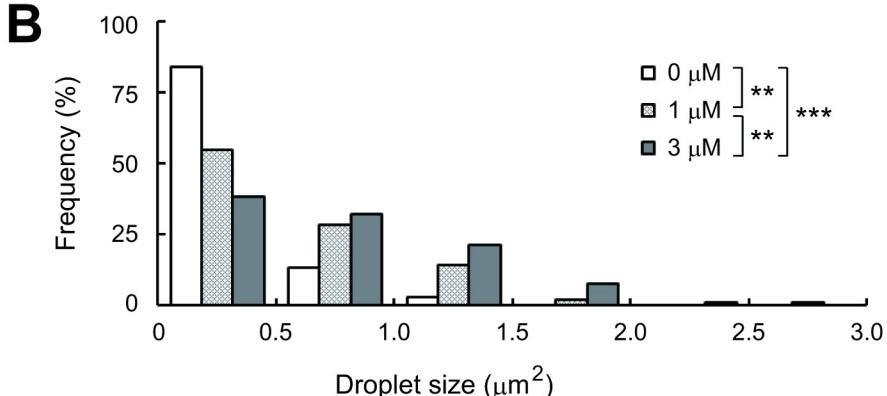

**Fig 8. FAM81A facilitates LLPS of postsynaptic proteins in vitro.** (A) Confocal microscopic images of LLPS of FAM81A with PSD-95, GluN2B carboxyl tail, and SynGAP1. iFluor 488-labeled PSD-95, NirFP-labeled GluN2B, and iFluor 405-labeled SynGAP1 (3 μm each) were mixed with increasing concentrations (0, 1, and 3 μm) of iFluor 568-labeled FAM81A. Scale bars: 5 μm. (B) The histogram of droplet size distribution. $^{**}P < 0.01$, $^{***}P < 0.001$ by one-way analysis of variance (ANOVA) followed by the Tukey–Kramer test. LLPS, liquid–liquid phase separation; PSD, postsynaptic density.

profiles with FAM81A expressed exclusively in brain and FAM81B in testis. Because the FAM81 ortholog in zebrafish is not expressed in synapses [19], it is likely that FAM81A evolved the capacity for expression and interaction with synaptic proteins in the mammalian lineage.

LLPS is a phenomenon in which proteins within a cell or subcellular compartment undergo a liquid–liquid phase transition, leading to the formation of distinct liquid-like or gel-like compartments within the cell. These compartments are often referred to as membrane-less organelles or biomolecular condensates. We found that FAM81A undergoes LLPS forming condensates with PSD-95, SynGAP, and GluN2B, which are key proteins controlling synaptic plasticity. Synaptic plasticity involves the modulation of synapse size. We found that FAM81A facilitates condensate formation in a dose-dependent manner (Fig 8) and that modulation of FAM81A expression resulted in a change in the size of dendritic spines (Fig 6A and 6B) and

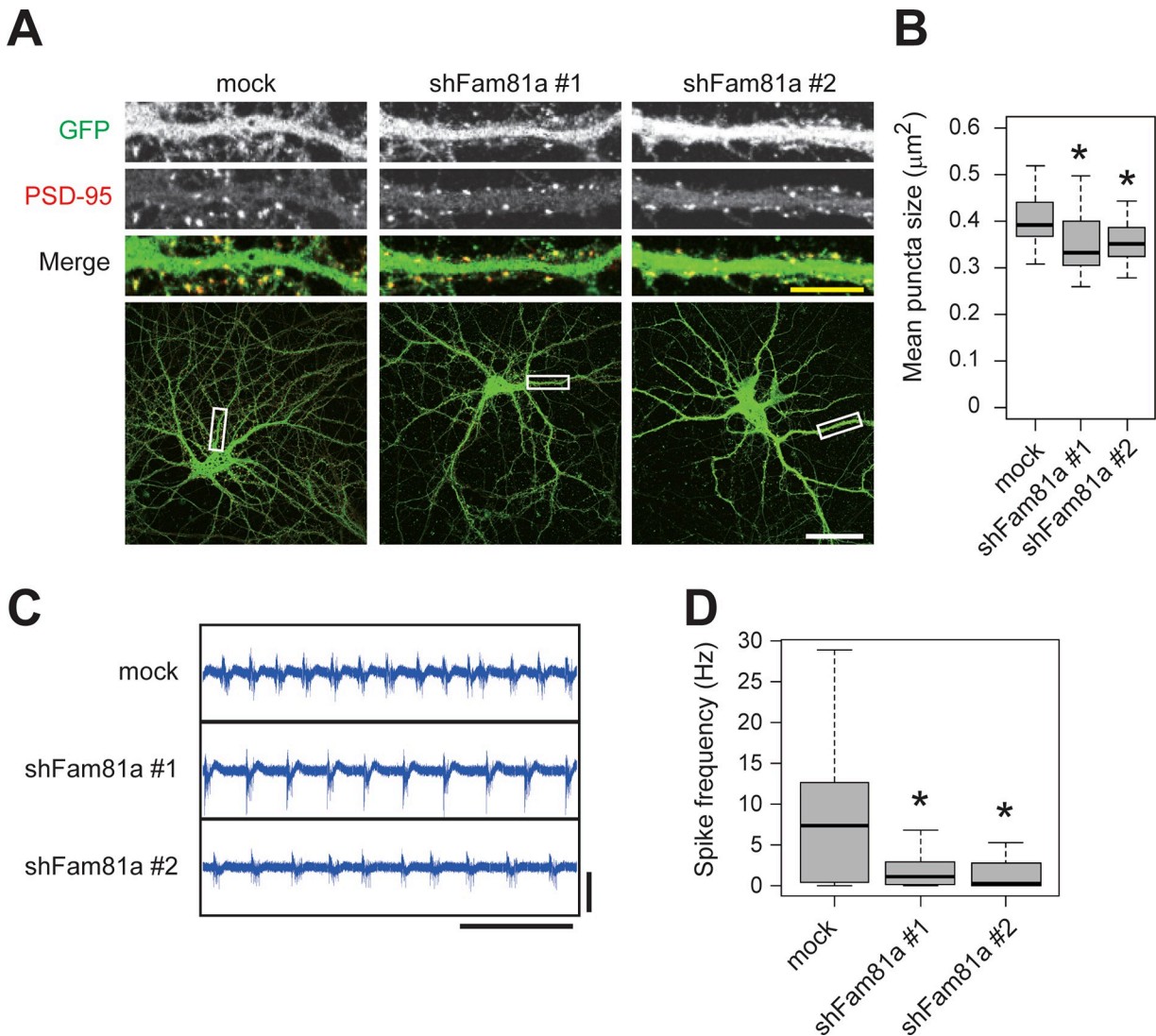

**Fig 9. FAM81A modulates PSD size and neuronal activity.** Primary cultured mouse hippocampal neurons were infected with lentivirus-encoding FAM81A shRNAs at DIV14. At DIV21, the neurons were subjected to immunocytochemistry (A and B) or electrophysiological recording using the MEA (C and D). (A and B) Neurons were fixed and labeled with anti-PSD-95 antibody. Representative dendrite images (A) and the mean size of PSD-95 puncta (B) are described. The numbers of analyzed neurons is 51 (mock), 47 (shFAM81A #1), and 49 (shFAM81A #2), respectively. (C and D) Neurons were subjected to extracellular electrophysiological recording using MEA. Representative signals (C) and spike frequency (D) are described. Scale bars: 50 μm (A, white), 10 μm (A, yellow), 10 s (C, x-axis), and 100 μV (C, y-axis). *$P < 0.05$. DIV, day in vitro; MEA, multielectrode array; PSD, postsynaptic density.

PSD-95 puncta (Fig 9A and 9B). These findings suggest that the interactions of FAM81A with other synaptic proteins within the condensates is important for synaptic plasticity. In addition, the protein–protein interactions and LLPS of FAM81A with these major postsynaptic molecules might be involved in the effect of FAM81A on neuronal activity (Fig 9C and 9D), as it has been proposed that synaptic activities are modulated through protein–protein interaction and LLPS in PSD [51–55] and our findings now indicate that FAM81A is directly involved with these same mechanisms.

Multivalent protein–protein interactions and intrinsically disordered regions (IDRs) are 2 important factors that drive LLPS [56]. We found that the IDR of FAM81A in the C-terminal

half of the protein was not essential for LLPS (Fig 5). Instead, we found the region between the 2 coiled-coiled domains (107–157) and the N-terminal domain (1–36) were critical for LLPS, possibly by impairments in their multivalent protein–protein interactions.

In addition to synaptic molecules, there are several major neuronal proteins that undergo LLPS, including TDP-43, FMRP, and CTTNBP2 [57–59]. Time-lapse imaging of FAM81A condensates in the cytoplasm of hippocampal neurons and HEK293T cells revealed dynamic changes that were similar to those observed with TLS/FUS [60], which is the first identified and best characterized RNA-binding proteins in the field of phase separation [59]. Although we currently have no direct evidence that FAM81A condensates contain RNA granules, FAM81A may play a role in signaling between synapses and dendrites. Shaft-localized condensates of FAM81A suggest that they may regulate local translation in response to various stimuli, including synaptic transmission [58,61] like TLS/FUS [62] or FMRP [63] in addition to synaptic function.

A number of genetic studies point to the genes encoding excitatory synaptic proteins as genes related to neuropsychiatric disorders such as schizophrenia and autism. We found that common variants of FAM81A have been registered in GWAS Catalog [64]. These variants of FAM81A (rs28890483 and rs10519005) are located at the 5′ side of the FAM81A gene, which may affect the expression level of FAM81A. They might increase the risk of bipolar disorder and schizophrenia (rs28890483 P = $9 \times 10^{-6}$, OR = 1.3904) or alcohol dependence (rs10519005 P = $5 \times 10^{-6}$, OR = 1.26). In addition, FAM81A is reported as one of the hub genes related to susceptibility to depression [65], suggesting that the expression level of FAM81A, which may affect neuronal activity, is involved in neuropsychiatric disorders, including major depression.

In conclusion, in addition to revealing novel properties of FAM81A, our results provide a large list of PSD proteins, which after characterization may also reveal important structural and functional roles at the synapse.

## Methods

### PSD proteome datasets and meta-analysis

Thirty-five datasets published in the following papers were referred [9–18,21,22,24–30,47,66,67]. Datasets composed of less than 30 proteins were not used. The protein ID described in each report was converted to mouse Entrez Gene ID using db2db of bioDBnet [68]. Proteins that failed to ID conversion were eliminated from the list. Heatmaps were generated in R software using the heatmap.2 function in the gplots package.

### Mice

The animal experiments performed in Japan were approved by RIKEN Animal Research Committee (Permission number: W2019-2-42) and Kobe University Institutional Animal Care and Use Committee (Permission number: A220510) and carried out in accordance with the institutional regulations and the guidelines by Science Council of Japan. The animal procedures conducted in the UK were approved by Edinburgh University Director of Biological Services and performed in accordance with the Animal (Scientific Procedures) Act 1986, UK Home Office regulations (Project license: PF3F251A9). ICR mice purchased from Japan SLC or C57BL/6J mice in the University of Edinburgh were used for the experiments. After cervical dislocation, the brain was removed from mice, briefly rinsed with ice-cold HBSS (Hanks' balanced salt solution), frozen with liquid nitrogen, and stored at −80°C before use for biochemical experiments.

## Preparation of PSD fraction

Preparation of the PSD-I fraction was performed according to the previously described protocol with minor modification [4]. Briefly, 3 brains obtained from adult (12-week-old) ICR mice were homogenized with a glass-Teflon homogenizer in Solution A (0.32 M sucrose, 1 mM NaHCO$_3$, 1 mM MgCl$_2$, 0.5 mM CaCl$_2$, and cOmplete EDTA-free Protease Inhibitor Cocktail). The homogenate was centrifuged at 1,400 g for 10 min at 4˚C to obtain the pellet and the supernatant. The pellet was resuspended in Solution A and centrifuged at 700 g for 10 min at 4˚C. The supernatant of the first and second centrifugation was pooled as an S1 fraction and subjected to subsequent centrifugation at 13,800 g for 10 min at 4˚C. The resulting pellet (P2 fraction) was resuspended with Solution B (0.32 M Sucrose and 1 mM NaHCO$_3$) and centrifuged in a sucrose density gradient (0.85/1.0/1.2 M) for 2 h at 82,500 g. Synaptosomes were collected from the 1.0/1.2 M border and diluted twice with Solution B. The synaptosome was lysed by adding an equal volume of solution C (1% TX-100, 0.32 M Sucrose, 12 mM Tris-HCl (pH 8.1)) and rotation at 4˚C for 15 min. The sample was centrifuged at 32,800 g for 20 min at 4˚C to obtain PSD-I as a resulting pellet.

## Plasmids

pCI-EGFP-NR2b wt (GFP-GluN2B) (Plasmid #45447), EBFP2-N1 (BFP) (Plasmid #54595), pLKO.1 - TRC cloning vector (Plasmid #10878), psPAX2 (Plasmid #12260), and pMD2.G (Plasmid #12259) were obtained from Addgene. GFP-SynGAP was gifted by Yoichi Araki and Richard Huganir (Johns Hopkins University). MG3C-SynGAP (Zeng and colleagues) was gifted by Mingjie Zhang (Southern University of Science and Technology). pβActin empty vector, pβActin-DsRed (DsRed), and pβActin-PSD-95-GFP (PSD-95-GFP) were gifted from Shigeo Okabe (The University of Tokyo). Full-length FAM81A cDNA (NM_029784.2) was cloned from the cDNA of the mouse brain and inserted into pEGFP-C1 (GFP-FAM81A) and pEGFP-N1 (FAM81A-GFP). To construct GST-FAM81A, FAM81A cDNA was inserted into pGEX-6P-3. To construct FAM81A-FLAG, FAM81A cDNA and 3xFLAG sequence were inserted into the pβActin vector. Truncated mutants of FAM81A were constructed by inverse PCR or amplification of target sequence by PCR. To construct PSD-95-mCherry, rat PSD-95 sequence (amplified from pβActin-PSD-95-GFP) and mCherry sequence were inserted into the pβActin vector. To construct pLKO.1-GFP, the puromycin-resistant gene was removed, and the EGFP sequence was inserted into the same site. To construct shRNA plasmids, shRNA sequences were designed using Invitrogen Block-iT RNAi Designer (https://rnaidesigner. thermofisher.com/rnaiexpress/). The oligonucleotides were inserted into pLKO.1 - TRC cloning vector and pLKO.1-GFP, according to the instruction (http://www.addgene.org/protocols/ plko/). Sequences of the oligonucleotides are as follows: shFam81a #1 F: CCGGgcaactgaatcggg atattgaCTCGAGtcaatatcccgattcagttgcTTTTTG, R: AATTCAAAAAgcaactgaatcgggatattgaCTC GAGtcaatatcccgattcagttgc. shFam81a #2 F: CCGGgctcctggacactaaatttaaCTCGAGttaaatttagtgt ccaggagcTTTTTG. R: AATTCAAAAAgctcctggacactaaatttaaCTCGAGttaaatttagtgtccaggagc.

## Cell culture

HEK293T cells and Lenti-X 293T cells (Takara, 632180) were cultured using a regular medium (Dulbecco's Modified Eagle Medium (DMEM; Nacalai Tesque, 08458–45), 10% fetal bovine serum (FBS; Gibco, 10270), and penicillin/streptomycin (Nacalai Tesque, 26253–84)) in a 5% CO$_2$ incubator. For 1,6-hexanediol treatment, a regular medium containing 10% 1,6-hexanediol (Sigma-Aldrich, 240117-50G) was used.

## Primary culture of hippocampal neurons

Hippocampi were dissected from E16.5 mouse embryos and dissociated using Neuron Dissociation Solutions (Wako, 291–78001). Neurons were counted using TC20 Automated Cell Counter (Bio-Rad) and then plated onto 24-well plates with coverslips (Matsunami, C013001), 60-mm dishes, 35-mm glass bottom dishes (Matsunami, D11130H) or 6-well MEA dishes (Multi Channel Systems, 60-6wellMEA200/30iR-Ti-tcr), which are previously coated with 0.01% poly-L-lysine in 0.1 M borate buffer solution; $4 \times 10^4$ cells (for 24-well plates), $4 \times 10^4$ cells (for 60-mm dishes), $1.8 \times 10^5$ cells (for 35-mm glass bottom dishes), or $1.6 \times 10^5$ cells (for MEA dishes) were plated in plating media (neuron culture media plus 5% FBS). After 2 to 14 h, the media was changed to neuron culture media (Neurobasal (Thermo Fisher Scientific, 21103049), 1× B-27 Supplement (Thermo Fisher Scientific, 17504–044), 1× GlutaMAX-I (Thermo Fisher Scientific, 35050–061), and penicillin-streptomycin (Nacalai Tesque, 26253–84)). At DIV4, D,L-(-)-2-amino-5-phosphovaleric acid (D,L-APV; SIGMA, A5282) was added to the media with a final concentration of 200 μm. Subsequently, half of the medium was changed once (for 24-well plate and 60-mm dishes) or twice a week (for MEA dishes).

## Live imaging of hippocampal neurons

Cells seeded on 35-mm glass bottom dishes (Matsunami, D11130H) were observed using Cell-Voyager CV1000 (Yokogawa Electric) equipped with a 60× objective lens. During live imaging, the culture dish was placed in a chamber to maintain incubation conditions at 37°C with 5% $CO_2$. Two-color time-lapse images were acquired at 30 min or 5-s intervals for hippocampal neurons or HEK293T cells, respectively. In the observation of hippocampal neurons, a 20 μm range of Z-stack images (21 slices, 2 μm) were acquired. As for HEK293T cells, a 2 μm range of Z-stack images (3 slices, 1 μm) were acquired. Maximum intensity projection images were shown in the figures and the movies.

## Immunoprecipitation

For immunoprecipitation of overexpressed FAM81A, HEK293T cells cultured on 10 cm dishes at approximately 50% confluency were transfected with 7.5 μg of FAM81A-FLAG and 7.5 μg of GFP tagged PSD-95, SynGAP, or GluN2B using PEI Max (Polysciences, 24765–1), and 24 h after transfection, cells were washed with ice-cold PBS, collected with centrifugation (5,000 rpm, 2 min 4°C), and resuspended with lysis buffer (1% Triton X-100, 50 mM Tris-HCl (pH 7.4), 150 mM NaCl, 1 mM EDTA, 15 mM NaF, 2.5 mM $Na_3VO_4$ with cOmplete EDTA-free Protease Inhibitor Cocktail). Lysates were kept on ice for 10 min and then centrifuged with 15,000 g for 15 min at 4°C. The supernatant was subjected to immunoprecipitation using 20 μl of ANTI-FLAG M2 Affinity Gel (Sigma-Aldrich, A2220-5ML). After 90 min incubation at 4°C, the samples were washed 5 times using lysis buffer, resuspended with sample buffer (62.5 mM Tris-HCl (pH 6.8), 4% sodium dodecyl sulfate (SDS), 10% glycerol, 0.008% bromophenol blue, and 25 mM dithiothreitol), and then boiled for 5 min. For immunoprecipitation of endogenous PSD-95, the P2 fraction was first prepared as described above from adult mouse forebrain. The pellet was resuspended with lysis buffer containing 1% sodium deoxycholate (DOC) and 0.4% SDS. The lysate was kept on ice for 10 min and then incubated with 2 anti-PSD-95 antibodies (NeuroMab, 75–028) or control IgG (Santa Cruz, sc-2025) at 4°C, for 90 min. The samples were then mixed with Protein G Sepharose beads (Amersham, 17-0618-01) and incubated at 4°C, for 30 min. The samples were then washed 5 times using lysis buffer containing 1% sodium DOC and 0.4% SDS, resuspended with sample buffer, and boiled for 5 min.

## Immunoblotting

In most experiments, the samples were subjected to SDS-polyacrylamide gel electrophoresis (SDS-PAGE) together with a molecular weight marker (Bio-Rad, 1610373). Proteins were transferred onto Immobilon-FL polyvinylidene difluoride membranes (Millipore, IPFL00010). The membrane was blocked in blocking buffer (Tris-buffered saline with 0.1% Tween 20 (TBST) with 5% skim milk) at room temperature and then incubated with the indicated primary antibody overnight in blocking buffer at 4˚C. The membrane was washed 3 times with TBST, followed by incubation with the respective secondary antibody in a blocking buffer for 1 h at room temperature. The membrane was washed 5 times with TBST. Immobilon Crescendo (Millipore WBLUR0500) or Chemi-Lumi One Super (Nacalai 02230–14) was used as a chemiluminescence substrate. The signals were analyzed using ImageQuant800 (AMERSHAM). Mouse anti-FLAG antibody (SIGMA, F3165), rabbit anti-GFP antibody (Thermo Fisher Scientific, A-6455), rabbit anti-FAM81A antibody Ab1 (Novus, NBP2-33295), rabbit anti-PSD-95 antibody (Abcam, ab18258), rabbit anti-synaptophysin antibody (Cell Signaling Technology, #4329), or mouse anti-β-actin antibody (SIGMA, A1978), were used as primary antibodies. Peroxidase goat anti-rabbit IgG (Jackson Immuno Research, 111-035-003) or HRP anti-mouse IgG (Amersham, NA9310) were used as secondary antibodies. For the experiment of endogenous PSD-95 immunoprecipitation (Fig 3B), the following reagents and apparatus were used: molecular weight marker (NEB, P7719S), mouse anti-PSD-95 antibody (NeuroMab, 75–028), rabbit anti-FAM81A antibody Ab2 (MERCK, HPA065797), SuperSignal West Femto (Thermo Fisher Scientific, 34096), ODYSSEY Fc (LI-COR). Anti-FAM81A antibody Ab1 is raised for amino acids 86–225 of human FAM81A and Ab2 is raised for amino acids 262–368. Consistent with the previous report [43], both antibodies detected a major band above 40 kDa in the mouse brain lysate (Fig 3A and 3B), at the expected size of FAM81A protein, suggesting that the endogenous FAM81A is successfully detected by these antibodies.

## Immunocytochemistry and confocal fluorescence microscopy

HEK293T cells or hippocampal neurons grown on coverslips were washed with PBS and fixed at room temperature using 4% paraformaldehyde (PFA) in PBS for 10 min or 4% PFA and 4% sucrose in PBS for 15 min, respectively. For immunocytochemistry of endogenous PSD-95, hippocampal neurons were incubated with blocking solution (2% normal goat serum (NGS), 0.2% Triton X-100, PBS) for 1 h at room temperature. Cells were washed with PBS twice and then incubated in antibody solution (2% NGS, 0.2% Triton X-100, PBS) with anti-PSD-95 antibody (Millipore, MABN68) for overnight at 4˚C. Cells were washed with PBS 3 times and incubated with antibody solution with Alexa Fluor 568-conjugated anti-mouse IgG antibody (Molecular Probes, A-11019) for 1 h at room temperature. Cells were washed with PBS 4 times and embedded with Mounting Medium (Vectashield, H-1000). Cells were observed using FV3000 confocal microscopy (Olympus) equipped with a 40× objective lens (Olympus, N2246700), a 60× objective lens (Olympus, N1480700), or a 100× objective lens (Olympus, N5203100). Immersion oil (Olympus, IMMOIL-F30CC) or silicone immersion oil (Olympus, SIL300CS-30CC) were used for 60× and 100× lenses, respectively.

## Immunohistochemistry

Mice were fully anesthetized by intraperitoneal injection of pentobarbital and transcardially perfused with PBS, followed by PBS with 4% PFA. Brains were dissected and incubated in PBS with 4% PFA at 4˚C for 3 h, then in PBS with 30% sucrose solution, and at 4˚C for 48 to 72 h. The samples were embedded with an OCT embedding matrix (CellPath, KMA-0100-00A) inside a plastic mold (Sigma-Aldrich, E6032-1CS), placed in beakers containing isopentane,

and frozen on liquid nitrogen. Frozen brains were stored at −80˚C until use. Frozen brain samples were cut at 18 μm thickness using a cryostat (Leica CM3050 S) to obtain sagittal sections. Cut brain sections were placed on Superfrost Plus glass slides (Epredia, J1800AMNZ) and dried up at room temperature overnight in the dark. After incubation in PBS for 10 min, the sections were incubated in Tris-buffered saline (TBS) with 5% bovine serum albumin (BSA) and 0.5% Triton X-100 for 1 h at room temperature. Sections were then incubated with rabbit anti-FAM81A antibody (MERCK, HPA065797) or mouse anti-GluN1 antibody (NeuroMab, 75–272) diluted in TBS with 3% BSA and 0.5% Triton X-100 at 4˚C for overnight. After washing 3 times with TBS with 0.5% Triton X-100, the sections were incubated for 2 h with Alexa Fluor 568-conjugated anti-rabbit IgG antibody (Thermo Fisher Scientific, A-10040) or Alexa Fluor 488-conjugated anti-mouse IgG1 antibody (Thermo Fisher Scientific, A-21121). Sections were then washed 3 times with TBS with 0.5% Triton X-100, mounted using MOWIOL 4–88 (Calbiochem, 475904) solution, and covered with a coverslip (VWR, 631–0153). Sections were observed using ECLIPSE Ti2 confocal microscopy (Nikon) equipped with a 4× objective lens (Nikon, MRD00045) or a 100× objective lens (Nikon, MRD01902). Immersion oil (ZEISS, Immersol 518 F) was used for the 100× lens.

### Preparation of recombinant FAM81A protein

Preparation of recombinant FAM81A protein was performed as follows. *Escherichia coli* strain BL21(DE3)-codonPlus-RIL (Aligent, 71136) was transformed with GST-FAM81A. Bacteria were cultured until OD600 reached approximately 0.8, and then isopropyl β-D-1-thiogalacto-pyranoside (IPTG) was added (final 0.03 mM). They were further cultured at 16˚C for 18 h, centrifuged at 1,800 g for 30 min at 4˚C, and then lysed with an ultrasonicator on ice (lysis buffer: 50 mM Tris (pH 8.0), 1,000 mM NaCl, 10 mM imidazole, 3 mM 2-mercaptoethanol, 10% glycerol, 50 mM L-arginine, 50 mM, L-glutamic acid, 10 mM betaine, 5% trehalose, and 0.2 mM PMSF). The sample was centrifuged at 100,000 g for 60 min at 4˚C. GST-FAM81A was pulled down using glutathione agarose beads (GST-accept, Nacalai-Tesque) and eluted with reduced glutathione (Nacalai Tesque, 17050–14) in the lysis buffer. While dialyzing in a buffer (25 mM Tris (pH 8.0), 100 mM NaCl, 5 mM DTT, 10% glycerol, 50 mM L-arginine, 50 mM L-glutamic acid, 10 mM betaine, 2.5% trehalose), 3C protease was added to remove GST, and then the cleaved GST tag was separated by subtractive glutathione agarose and purified in a size exclusion column buffer containing 25 mM HEPES (pH 8.0), 100 mM NaCl, 0.5 mM Tris-2-carboxyethyl phosphine (TCEP) using HiLoad 26/600 Superdex 200 pg size exclusion column. Fractions (>95% purity) were pooled, concentrated, aliquoted, and flash frozen in liquid nitrogen and stored at −80˚C until needed. Recombinant PSD-95, SynGAP, and GluN2B were prepared as previously described [49,53].

### Labeling and observation of LLPS of purified proteins

The FAM81A protein was labeled by iFluor 488- or iFluor 568-succinimidyl ester (AAT Bioquest) as previously described [53]. The PSD proteins were diluted in a phase buffer (50 mM Tris (pH 8.0), 100 mM NaCl, 1 mM TCEP, 0.5 mM EGTA, 5 mM $MgCl_2$, and 2.5 mM ATP). A protein mixture (5 μl) was injected into a homemade imaging chamber and observed by confocal microscopy (FLUOVIEW FV1200, Olympus). The number and size of the iFluor488-positive droplets were analyzed using Analyze Particles function of Fiji software.

### Lentivirus production and infection

Lenti-X 293T cells were co-transfected with pLKO.1 lentiviral plasmid, psPAX2, and pMD2.G using Lipofectamine LTX and PLUS Reagents (Thermo Fisher Scientific, 15338–100). After

overnight incubation, the medium was changed to a fresh medium, and 60 to 72 h after transfection, the supernatant was collected and filtrated through a 0.45 μm filter (Millipore, SLHV033RS). The virus was concentrated using Lenti-X Concentrator (Clontech, 631231), resuspended by one-twentieth volume of PBS, and stored at −80°C before use. For infection of cultured neurons, one-hundredth volume of virus solution was added to the neuron culture medium, and 8 h after infection, the medium was changed to fresh medium. For mock infection, pLKO.1 empty vector was used as a lentiviral plasmid.

## Real-time PCR

RNA was extracted from cortical neurons on 6 cm dishes using RNeasy Plus Mini Kit (QIAGEN, 74134) and reverse-transcribed using SuperScript II (Thermo Fisher Scientific, 18064–022). Real-time PCR was performed using the primers (Fam81a: F cttagccaggctgttcttgg, R ccagcgtctttaaggcagaa, Actb: F cgtgcgtgacatcaaagagaa, R tggatgccacaggattccat), Power SYBR Green PCR Master Mix (Thermo Fisher Scientific, 4367659), and QuantStudio3 (Thermo Fisher Scientific).

## Quantitative image analysis

The 1,024 × 1,024 pixel images obtained with confocal microscopy were analyzed using ImageJ. For quantification of FAM81A-GFP-positive structures in HEK293T cells, the signals were extracted by binarization using Find Maxima (noise tolerance: 120). The number and size of the signals in each cell were analyzed using Analyze Particles after selecting individual cells. Unhealthy (shrank) cells, cells with too high (saturated) or too low (invisible) signal intensity, and aggregated cells with unclear borders were avoided from the analysis. Images were first shuffled for blinded analysis to quantify PSD-95-positive structures in neurons expressing GFP. Then, images with single neurons were selected. PSD-95-positive structures were extracted by binarization using Find Maxima (noise tolerance: 120). Structures with more than 50 pixels were eliminated to avoid the detention of non-synaptic structures. To assess dendrite length, signals of GFP were traced using NeuronJ [69]. The PSD-95 puncta along the dendrites (within 25-pixel diameter) were extracted using SynapCountJ [70]. The mean puncta size and the number of puncta per dendrite length were calculated for individual neurons. The result was visualized as a boxplot using R software. Student *t* test was used for statistical analysis.

## Multielectrode array (MEA)

Hippocampal neurons plated in each well of a 6-well MEA dish were cultured until DIV21. The medium was exchanged with a fresh neuron culture medium without D,L-APV, and 30 min after the medium exchange, neuronal activity was analyzed using MEA2100 (Multi Channel Systems) and MC_Rack Version 4.6.2. Input voltage range and sampling frequency were set as ±19.5 mV and 20,000 Hz, respectively. Recording of the neuronal activity was performed for 2 min and repeated for 3 times. Voltage over 5 standard deviations was used as a threshold to detect spikes. The data of electrodes that do not detect any spikes through 3 replicates were not used for the analysis.

## Databases and bioinformatics tools

DAVID (https://david.ncifcrf.gov/) [71] and SynGO (https://www.syngoportal.org/) [72] were used to analyze the enrichment of synaptic proteins in the subsets of PSD proteins. NCBI Gene (https://www.ncbi.nlm.nih.gov/gene/) was referred to check gene expression in humans

and mice. Xenbase (http://www.xenbase.org/entry/) [73] was referred to check gene expression patterns in the frog. Protein BLAST (https://blast.ncbi.nlm.nih.gov/Blast.cgi) was used to identify orthologs of proteins. Gene2Function (https://www.gene2function.org/) [44] was also referred to find orthologs in invertebrates and unicellular organisms. SMART (http://smart.embl-heidelberg.de/) [74] was used to analyze domain architecture. D2P2 (http://d2p2.pro/) [75] was used to evaluate disordered regions. ConSurf [76] was used to assess evolutionary conservation. MARRVEL (http://marrvel.org/) [77] was referred to search for rare variants. GWAS Catalog (https://www.ebi.ac.uk/gwas/home) [64] was referred to search common variants.

### Mass spectrometry data analysis

Raw LC-MS/MS files (PRIDE partner repository dataset identifier PXD005630) from our previous proteomic study [19] of the composition of mouse synaptosome and PSD fractions were reprocessed using MaxQuant version 1.6.10.43 [25]. Data were searched against a human UniProt reference proteome (downloaded May 2020) using the following search parameters: enzyme set to Trypsin/P (2 mis-cleavages), methionine oxidation and N-terminal protein acetylation as variable modifications, cysteine carbamidomethylation as a fixed modification. A protein FDR of 0.01 and a peptide FDR of 0.01 were used for identification level cut-offs based on a decoy database searching strategy. Label-free quantification (LFQ) was performed and iBAQ values were calculated for proteins identified in synaptosome and PSD fractions, which allowed proteins to be ranked by their estimated relative abundance.

### Supporting information

**S1 Fig. Proteins detected in multiple datasets include core PSD proteins.** List of the top 27 proteins that show the highest number of datasets. Blue and yellow bars indicate the dataset number of unbiased and candidate-based approaches. Proteins detected in at least 26 datasets are listed.
(PDF)

**S2 Fig. Domain architecture of FAM81A/FAM81A homologs.** Coiled-coil regions detected by SMART and disordered regions predicted by IUPred2A.
(PDF)

**S3 Fig. Gene expression pattern of FAM81A and FAM81B in mouse and frog.** (A and B) Gene expression pattern of FAM81A and FAM81B in mouse (A) and frog (B) tissue. Mouse data and frog data were obtained from NCBI Gene and Xenbase, respectively. RPKM: Reads per Kilobase of exon per Million mapped reads; TPM: Transcripts per Kilobase Million.
(PDF)

**S4 Fig. Distribution of FAM81A at postnatal day 7 mouse brain.** (A and B) Distribution of FAM81A in brain. Immunohistochemistry of FAM81A was performed on sagittal section of postnatal day 7 mouse brain. Scale bar: 2 mm (A) and 10 μm (B).
(PDF)

**S5 Fig. Protein sequences of FAM81A orthologs in mammals, birds, and reptiles.** The sequence of FAM81A homologs of indicated species are aligned using Kalign and visualized using MView. The numbers on the sequences indicate the amino acid number of mouse FAM81A. Residues identical to human FAM81A are highlighted. The colors of the characters represent the classification of amino acids: hydrophobic (light green), large hydrophobic (dark green), positive (red), small alcohol (light blue), and polar (purple). Mammals: human (*Homo*

sapiens), rat (*Rattus norvegicus*), and mouse (*Mus musculus*). Birds: chicken (*Gallus gallus*) and zebra finch (*Taeniopygia guttata*). Reptiles: soft-shelled turtle (*Pelodiscus sinensis*) and gecko (*Paroedura picta*). The coiled-coil domains (CC1 and CC2) and low complexity region (LCR) of mouse FAM81A are labeled with a black bar.
(PDF)

**S6 Fig. Expression check of GFP-tagged FAM81A mutants in HEK293T cells.** (A) HEK293T cells were transfected with FAM81A-GFP or its mutant 24 h later, cells were lysed and protein concentration was quantified, and 30 μg protein was subjected to immunoblotting using anti-GFP or anti-β-actin antibodies. (B) Relative band intensity of panel A data quantified using ImageJ. $^*P < 0.05$, unpaired Student's *t* test.
(PDF)

**S7 Fig. Condensates of PSD-95 and SynGAP.** HEK293T cells were transfected with indicated plasmids, and 24 h later, cells were fixed and observed with confocal microscopy. Scale bars: 20 μm (white) or 4 μm (yellow).
(PDF)

**S8 Fig. Knockdown of FAM81A with lentivirus shRNA.** (A) Primary cultured mouse cortical neurons were infected with lentivirus encoding FAM81A shRNAs at DIV4. At DIV16, neurons were harvested to extract mRNA. After the preparation of cDNA, real-time PCR was performed. The relative expression level is described. (B) Primary cultured mouse hippocampal neurons were infected with lentivirus of pLKO.1-GFP mock plasmid at DIV14. At DIV21, the neurons were fixed and subjected to immunocytochemistry using an anti-PSD-95 antibody. Scale bars: 100 μm.
(PDF)

**S1 Table. List of 5,869 proteins included in PSD proteome datasets.** Proteins were described as mouse Entrez Gene ID, and 0 and 1 indicate undetected and detected, respectively.
(XLSX)

**S2 Table. List of 177 proteins in PSD proteome datasets that have not been fully characterized.**
(XLSX)

**S1 Movie. FAM81A droplets in hippocampal neuron.** Mouse primary hippocampal neurons were transfected with FAM81A-GFP and DsRed. Images were obtained every 30 min.
(MPEG)

**S2 Movie. FAM81A droplets in HEK293T cells.** HEK293T cells were transfected with FAM81A-GFP. Images were obtained every 5 s.
(MPEG)

**S3 Movie. Aggregation-like structure of FAM81A in HEK293T cells.** HEK293T cells were transfected with FAM81A-GFP. Images were obtained every 5 s.
(MPEG)

**S1 Data. Underlying data for Figs 1–6, 8, 9, S1, S3, S6 and S8.**
(XLSX)

**S1 Raw Images. Raw gel images for Figs 3, 6, 7 and S6.**
(PDF)

## Acknowledgments

We thank Y. Araki, R. Huganir, M. Zhang, and S. Okabe for providing the plasmids; and A. Okuda, K. Morishima, M. Sugiyama, T. Imasaki, and R. Nitta for helpful discussion. We thank T. Otani, C. Noguchi, S. Yoshida, S. Fujima, and J. Tharaux for their technical assistance and all technical staff in the Takumi lab for preparing experimental reagents. We thank J. Griffiths for proofreading of the manuscript.

## Author Contributions

**Conceptualization:** Takeshi Kaizuka.

**Data curation:** Takeshi Kaizuka.

**Formal analysis:** Takeshi Kaizuka.

**Funding acquisition:** Yasunori Hayashi, Toru Takumi.

**Investigation:** Takeshi Kaizuka, Taisei Hirouchi, Toshihiko Shirafuji.

**Methodology:** Takeshi Kaizuka, Takeo Saneyoshi, Mark O. Collins.

**Project administration:** Toru Takumi.

**Supervision:** Seth G. N. Grant, Yasunori Hayashi, Toru Takumi.

**Writing – original draft:** Takeshi Kaizuka.

**Writing – review & editing:** Seth G. N. Grant, Yasunori Hayashi, Toru Takumi.

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
