## [Editor Report · Decision Letter 0]

17 Jan 2023

Dear Dr Takumi, 

Thank you for submitting your manuscript entitled "DRACC1, a major postsynaptic protein, regulates the condensation of postsynaptic proteins via liquid-liquid phase separation" for consideration as a Research Article by PLOS Biology. I apologize for our delay in sending you an initial decision - we have been working through a bit of a backlog since going on break over the Holidays. 

Your manuscript has now been evaluated by the PLOS Biology editorial staff as well as by an academic editor with relevant expertise and I am writing to let you know that we would like to send your submission out for external peer review.

Once your full submission is complete, your paper will undergo a series of checks in preparation for peer review. After your manuscript has passed the checks it will be sent out for review. To provide the metadata for your submission, please Login to Editorial Manager (https://www.editorialmanager.com/pbiology) within two working days, i.e. by Jan 19 2023 11:59PM.

Kind regards,

Lucas

Lucas Smith, Ph.D.

Associate Editor

PLOS Biology

lsmith@plos.org

---

## [Decision Letter · Decision Letter 1]

28 Feb 2023

Dear Dr Takumi,

Thank you for your patience while your manuscript "DRACC1, a major postsynaptic protein, regulates the condensation of postsynaptic proteins via liquid-liquid phase separation" was peer-reviewed at PLOS Biology. Your manuscript has been evaluated by the PLOS Biology editors, an Academic Editor with relevant expertise, and by several independent reviewers.

As you will see in the reviewer reports, which can be found at the end of this email, although the reviewers find the work potentially interesting, they have also raised a substantial number of important concerns. We think that, in order to offer the strength of advance that we would require for publication at PLOS Biology, these concerns would need to be thoroughly addressed, with new data and analyses where appropriate.

After discussion with the Academic Editor we think the study would need to be expanded to test the expression of Fam81A, in vivo, as Reviewer 1 has suggested. Additionally, we feel that it would be important to complement the in vitro studies, by testing the function of Fam81A in synaptic transmission, in vivo (by for example using a virus-mediated knockout/knockdown/labeling approach). We also think that it would be important to address Reviewer 3's comments, and, if possible, to strengthen the meta-analysis performed here, as we think this would increase the value of this as a resource to the community.

As a note Reviewer 2 has raised concerns with the novelty of the study and whether the findings are a good fit for PLOS Biology, given that Fam81A has been observed in the PSD before. After discussion with the Academic Editor, we are less concerned about the conceptual advance of the study if the manuscript is developed further, as outlined above. We do, however, agree with Reviewer 2 that you should not change the name of Fam81A.

We do understand that these requests represent a large amount of work, and we are therefore willing to work with you by extending the deadline for your review and/or by providing input on a revision plan if you think that would be helpful. We would also understand if you do not wish to take on this additional work and if you wish to pursue publication elsewhere. If you would prefer, we would be willing to explore a possible transfer of a more modest revision to PLOS Genetics (the PLOS journals are editorially independent, and so I cannot guarantee an outcome of that discussion).

Given the extent of revision needed, we cannot make a decision about publication until we have seen the revised manuscript and your response to the reviewers' comments. Your revised manuscript is likely to be sent for further evaluation by all or a subset of the reviewers.

We expect to receive your revised manuscript within 3 months, although as mentioned, would be happy to extend this deadline if needed. Please email us (plosbiology@plos.org) if you have any questions or concerns, or would like to request an extension.

**IMPORTANT - SUBMITTING YOUR REVISION**

*Re-submission Checklist*

*Published Peer Review*

*PLOS Data Policy*

*Blot and Gel Data Policy*

Sincerely,

Lucas

Lucas Smith, Ph.D.

Associate Editor

PLOS Biology

lsmith@plos.org

REVIEWS:

Reviewer #1, Kihoon Han (note - Reviewer 1 has signed this review): In this manuscript, Kaizuka et al. used systematic approaches to identify uncharacterized PSD proteins from previous proteomic analyses. Among the 97 candidates, the authors further characterized the top-ranked protein FAM81A, which they renamed as DRACC1. Using biochemical, cell biological, and electrophysiological analyses, the authors provided evidence that DRACC1 can interact with some core PSD proteins and induce LLPS-mediated condensation of the proteins. This study provides a nice example of how to identify and understand the roles of previously uncharacterized PSD proteins, expanding our knowledge about the complexity and diversity of the PSD proteome. Additionally, the manuscript is well-written and describes key findings in a solid manner. I have a few comments that will strengthen the main conclusions of the manuscript:

1. Although the authors provide information about the conservation and expression profiles of DRACC1/2 in Figure 2 and Figure 3A, more direct validations of this information at the protein level should be performed. Additionally, detailed information about the antibody against DRACC1 (Figure 3A) needs to be included in the manuscript (e.g., specificity between DRACC1 and 2). Once the antibody is validated, brain regional and age-dependent expression, subcellular distribution, etc. can be analyzed with the antibody using at least mouse or rat brain samples.

2. In Figure 4, the authors characterized various deletion mutants of DRACC1 to identify critical regions for the formation of condensates. It is necessary to show whether those different deletion mutants have similar expression levels in cells by western blotting.

3. Regarding the formation of condensates by DRACC1, the experimental results suggest that the N-terminal regions are critical while the IDR is in the C-terminus. How can this be explained?

4. Regarding the protein interaction between DRACC1 and core PSD proteins (Figure 5), the interaction should be validated in vivo using PSD samples.

5. In addition, whether such interaction in HEK293T cells requires condensation of DRACC1 can be tested with the deletion constructs used in Figure 4.

6. In Figure 7, the effects of shDracc1 look subtle for both PSD-95 puncta size and spike frequency. Can other parameters, such as excitatory synapse density, dendritic spine size/morphology, etc. be analyzed? In addition, effects of overexpression of WT or deletion mutants of DRACC1 can be tested for the same parameters to further confirm the functional significance of condensation of DRACC1.

Reviewer #2: Here, Kaizuka et al describe the role of the PSD protein FAM81A. The manuscript focus in a meta-analysis of PSD (protein and PPI) datasets to select understudied PSD proteins. Unfortunately, in the way that is presented, the manuscript has very little new information to offer and in the current format is not suitable for publication.

It is not clear what the so-called "meta" analysis is, or how it was performed. It seems that is only looking at different datasets and check what proteins are more replicated but with not much information in literature. The authors selected FAM81A as their candidate, perform a series of assays and because of their findings they conclude that the "meta" analysis is a very important tool. However, it seems that this is a hand pick molecule, so I'm not sure what are they validating.

Unfortunately, the candidate selected, FAM81A, has been already characterized as a PSD protein including protein localization by EM. See PMID: 30735723. This makes the whole first part of the manuscript redundant and inaccurate in terms of novelty. The section describing FAM81A PPIs is also redundant as the authors clearly show that PPIs with several PSD proteins including PSD95 and SYNGAP1 have been already described in literature. The authors then show that FAM81A under certain experimental conditions can induce phase-separation when expressed with their protein interactors. This is expected as most of PPIs can induce phase-separation under the reported experimental conditions. In fact, phase-separation can be viewed as other way to observe PPI. This information is also redundant.

The authors provide two new pieces of information. One is that the n-terminal region of FAM81A might be necessary for PPI with the reported PSD proteins, and also that knockdown of FAM81A affects PSD size and neuronal activity. While this information is novel I consider it not enough for publication. The manuscript should need to be re-written and focus in these assays and remove what is known, at least not presented as new information.

SynGO should be used as database analysis instead of GO. 

The authors should not propose to change the protein id of FAM81A for DRACC1. This has no point and only will create problems with future annotations and databases. FAM81A is already reported as a PSD protein, and annotated as this in SynGO. I strongly oppose to a change in the protein id.

Reviewer #3: The manuscript by Kaizuka et al. describes a meta-analysis of 35 previously published PSD protein datasets, 20 of which are derived from mass spectrometry analysis of enriched PSD fractions and 15 of which are candidate-based using immunoprecipitation, affinity enrichment, or proximity labeling approaches. This meta-analysis could have been done better, nonetheless, it was effective enough to generate interesting candidate PSD proteins for further study. Assuming that proteins common to many datasets are more likely to be bona fide PSD proteins, the authors studied the function of a previously uncharacterized protein, Fam81 (renamed DRACC1) with regards to the PSD. They showed that DRACC1 promotes liquid-liquid phase separation of several major and well-characterized PSD proteins, and down-regulation of DRACC1 in neurons causes a decrease in PSD-95 puncta size and frequency of neuronal firing. Overall the manuscript is well-written, and the approach is a clever example of the utility of proteomic data meta-analysis. Characterization of DRACC1 furthers our understanding of the role of liquid-liquid phase separation in PSD formation and function. The manuscript should be useful for readers who are interested in the fields of PSD composition and function as well as liquid-liquid phase separation. 

Specific comments:

1. Page 2: Page 4, line 8: Please mention here that the authors have renamed this protein (until I read page 7 it was not clear who had renamed the protein)

2. last sentence on page 8 (first sentence on page 9): condensates found in both soma and dendritic shafts (Fig 3F, 3G)- it is not clear to me where soma are in Fig 3F and Fig 3G. (figures say PSD and dendritic shafts, respectively, nothing about soma). Do you mean PSD rather than soma? Next sentence, which condensates are moving rapidly compared to those in PSD? Movie S1 shows those in dendritic shaft are moving fast, is that what is meant here? Please clarify. 

3. page 13, lines 3-4: Estimates of the abundance of DRACC1 can be made from the mass spectrometry datasets. One or more high quality MS datasets could be analyzed either by spectral counting (how often are peptides from DRACC1 seen compared to peptides from other proteins, corrected by normalization based on number of theoretical tryptic peptides from DRACC1) or iBAQ (intensity-based quantitation, again normalized to number of theoretical tryptic peptides).

4. While the meta-analysis here was very useful for generating DRACC1 as a very interesting candidate, a better meta-analysis would have included re-analysis of the raw mass spectrometry files when available for each of the MS-based PSD profiles. This is most likely beyond the scope of the current manuscript but would improve the overall meta-analysis, especially since studies have shown that MS data analysis varies significantly between labs. 

5. The attrition rate due to failed ID conversion for proteins was very high in this manuscript. Again, re-analysis of raw MS data would help with this problem, but more attention to renaming the MS datasets for comparison would have been helpful even without direct analysis of MS data. Again, this may be beyond the scope of this manuscript, as the meta-analysis was only a starting point, not the main focus of the paper, and effective enough to ID DRACC1 as an interesting and important PSD component. 

Minor comments:

1. Page 3, line 5: remove "the" before "detergent"

2. Other minor grammatical errors remain throughout the manuscript

3. Figure 2C, "chicken" and "sea squirt" are misspelled

---

## [Decision Letter · Decision Letter 2]

5 Dec 2023

Dear Dr Takumi,

Thank you for your patience while we considered your revised manuscript "FAM81A, a major postsynaptic protein, regulates the condensation of postsynaptic proteins via liquid-liquid phase separation" for publication as a Research Article at PLOS Biology. This revised version of your manuscript has been evaluated by the PLOS Biology editors, the Academic Editor and by two of the original reviewers, all of whom are satisfied by the revision. 

While we are likely to accept this manuscript for publication, based on the reviewer comments, before we can editorially accept your study we need you to address a number of data and other policy-related requests and a revision that we think will not take very long.

**PLEASE ADDRESS THE FOLLOWING EDITORIAL REQUESTS: 

1) TITLE: We would like to suggest a minor change to the title to improve its flow. If you agree, we suggest you change it to: "FAM81A is a postsynaptic protein that regulates the condensation of postsynaptic proteins via liquid-liquid phase separation"

2) ABSTRACT: Please note that per journal policy, the model system/species studied should be clearly stated in the abstract of your manuscript. 

3) ETHICS STATEMENT: Please update the ethics statement in your methods section to include the full names of the IACUC/ethics committees that reviewed and approved the animal care and use protocol/permit/project license. Please also include an approval numbers. Please also update this statement to include the specific national or international regulations/guidelines to which your animal care and use protocols adhered. Please note that institutional or accreditation organization guidelines (such as AAALAC) do not meet this requirement.

4) BLURB: In the relevant section of our online system, please provide a blurb which (if accepted) will be included in our weekly and monthly Electronic Table of Contents, sent out to readers of PLOS Biology, and may be used to promote your article in social media. The blurb should be about 30-40 words long and is subject to editorial changes. It should, without exaggeration, entice people to read your manuscript. It should not be redundant with the title and should not contain acronyms or abbreviations.

5) WESTERN BLOTS: Please note that we require the original, uncropped and minimally adjusted images supporting all blot and gel results reported in an article's figures or Supporting Information files. We will require these files before a manuscript can be accepted so please prepare and upload them now. Please carefully read our guidelines for how to prepare and upload this data: https://journals.plos.org/plosbiology/s/figures#loc-blot-and-gel-reporting-requirements

6) CODE: Per journal policy, if any code was generated to support the conclusions of your manuscript, we would require that you make it available without restrictions upon publication. Please ensure that any code is sufficiently well documented and reusable, and that your Data Statement in the Editorial Manager submission system accurately describes where your code can be found.

7) DATA: You may be aware of the PLOS Data Policy, which requires that all data be made available without restriction: http://journals.plos.org/plosbiology/s/data-availability. For more information, please also see this editorial: http://dx.doi.org/10.1371/journal.pbio.1001797

a. Supplementary files (e.g., excel). Please ensure that all data files are uploaded as 'Supporting Information' and are invariably referred to (in the manuscript, figure legends, and the Description field when uploading your files) using the following format verbatim: S1 Data, S2 Data, etc. Multiple panels of a single or even several figures can be included as multiple sheets in one excel file that is saved using exactly the following convention: S1_Data.xlsx (using an underscore).

b. Deposition in a publicly available repository. Please also provide the accession code or a reviewer link so that we may view your data before publication. 

>>Regardless of the method selected, please ensure that you provide the individual numerical values that underlie the summary data displayed in the following figure panels as they are essential for readers to assess your analysis and to reproduce it:

Fig 1D,E; Fig 2A,D,E; Fig 3F-G; FIg 4D; Fig 5C-E; Fig 6B; FIg 8B; Fig 9B,D; Fig S1; Fig S3; Fig S6B; Fig S8

>>Please also ensure that figure legends in your manuscript include information on where the underlying data can be found, and ensure your supplemental data file/s has a legend.

>>Please ensure that your Data Statement in the submission system accurately describes where your data can be found.

We expect to receive your revised manuscript within two weeks. 

*Published Peer Review History*

*Press*

Sincerely,

Lucas

Lucas Smith, Ph.D.

Senior Editor,

lsmith@plos.org,

PLOS Biology

Reviewer remarks:

Reviewer #1, Kihoon Han (note, reviewer 1 has signed this review): In the revised manuscript, the authors performed a significant amount of improvement to address my concerns. I have no further comment.

Reviewer #3: The authors have responded adequately to my concerns. The manuscript reports a new function for FAM81A, a protein that was previously found in many proteomic studies of the PSD but whose function in the PSD was unknown. The authors show that FAM81A can facilitate increase in PSD95 punctate size, and its removal leads to decreased neuronal firing rates. This is consistent with a role in promotion of liquid-liquid phase separation by the PSD. This finding should be of interest to the readership of PLOS Biology.

---

## [Editor Report · Decision Letter 3]

17 Jan 2024

Dear Toru, 

Thank you for the submission of your revised Research Article "FAM81A is a postsynaptic protein that regulates the condensation of postsynaptic proteins via liquid-liquid phase separation" for publication in PLOS Biology and thank you also for addressing our last editorial requests in this revision. On behalf of my colleagues and the Academic Editor, Eunjoon Kim, I am pleased to say that we can in principle accept your manuscript for publication, provided you address any remaining formatting and reporting issues. These will be detailed in an email you should receive within 2-3 business days from our colleagues in the journal operations team; no action is required from you until then. Please note that we will not be able to formally accept your manuscript and schedule it for publication until you have completed any requested changes.

**IMPORTANT NOTES: 

1 - As discussed, I have updated the S1_raw western blot file with the fully annotated version that you provided me over email. I also updated your Figure S6 to reflect the version you provided me, with the correct orientation for the B-actin lanes. Please do double check that everything looks good with these changes. 

2 - Thank you for providing the data underlying your figures as a supplemental file (S1_data). I see that you have referenced this file in the text of your manuscript, which is fine, but generally unnecessary. So if you prefer, we would be OK if you removed the references to this file in the text. However, we do ask that you add a reference to this file in each relevant figure legend. For example, you can add a note at the end of each figure legend stating "the data underlying this figure can be found in S1_data". Please make this change as you address any formatting requests to come. 

PRESS

Sincerely, 

Luke

Lucas Smith, Ph.D., Ph.D.

Senior Editor

PLOS Biology

lsmith@plos.org